# XModBench: Benchmarking Cross-Modal Capabilities and Consistency in Omni-Language Models

**Xingrui Wang[1,2], Jiang Liu[1✉], Chao Huang[1,3], Xiaodong Yu[1], Ze Wang[1]**
**Ximeng Sun[1], Jialian Wu[1], Alan Yuille[2], Emad Barsoum[1], Zicheng Liu[1]**
[1]Advanced Micro Devices   [2]Johns Hopkins University   [3]University of Rochester

🌐 Project Page   🤗 Dataset Card   ⌗ Code

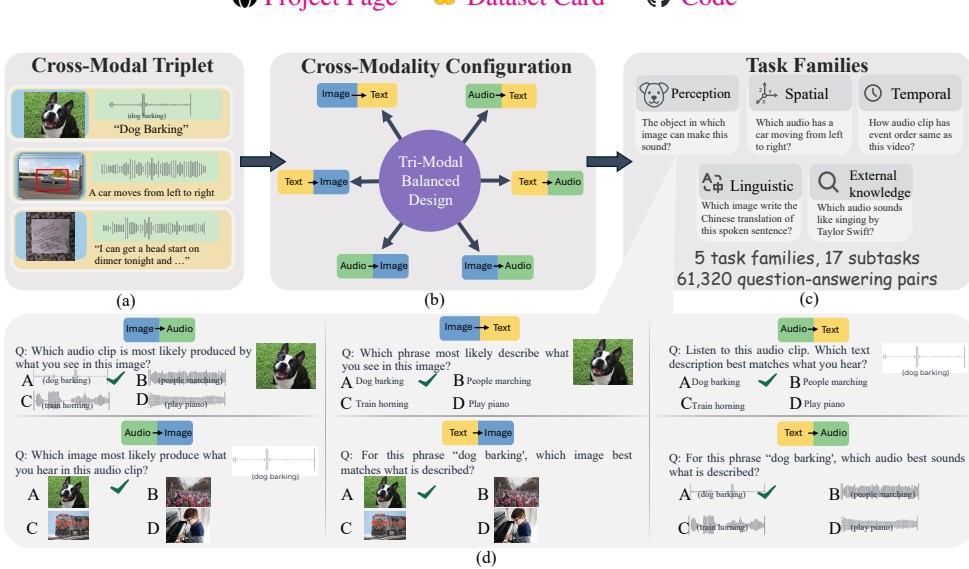

Figure 1: Overview of **XModBench**. (a) Instances are built from aligned text–image–audio triplets; (b) instantiated into six modality configurations by permuting context and candidate modalities; (c) spanning five domains with 17 subtasks and 61,320 question–answer pairs; and (d) illustrated with example multiple-choice questions under balanced modality settings.

## Abstract

Omni-modal large language models (OLLMs) aim to unify audio, vision, and text understanding within a single framework. While existing benchmarks primarily evaluate general cross-modal question-answering ability, it remains unclear whether OLLMs achieve modality-invariant reasoning or exhibit modality-specific biases. We introduce **XModBench**, a large-scale tri-modal benchmark explicitly designed to measure cross-modal consistency. XModBench comprises **61,320** multiple-choice questions spanning **five task families** and systematically covers all **six modality compositions** in question–answer pairs, enabling fine-grained diagnosis of an OLLM's modality-invariant reasoning, modality disparity, and directional imbalance. Experiments show that even the strongest model, Gemini 2.5 Pro, (i) struggles with spatial and temporal reasoning, achieving less than 60% accuracy, (ii) reveals persistent modality disparities, with performance dropping substantially when the same semantic content is conveyed through audio rather than text, and (iii) shows systematic directional imbalance, exhibiting lower consistency when vision serves as context compared to text. These findings indicate that current OLLMs remain far from truly modality-invariant reasoning, and position **XModBench** as a fundamental diagnostic tool for evaluating and improving cross-modal competence. All data and evaluation tools will be available at https://github.com/XingruiWang/XModBench.

---

✉ Corresponding author

# 1 INTRODUCTION

Omni-modal large language models (OLLMs) integrate text, vision, and audio into a unified reasoning framework (Comanici et al., 2025; Xu et al., 2025; Xing et al., 2025; Su et al., 2023; Fu et al., 2025b; Cheng et al., 2024; Zhong et al., 2025). However, despite impressive advancements and expanded modality coverage, a key question remains: do these models reason in a truly modality-invariant manner, or do they still exhibit systematic biases tied to specific input modalities? For humans, cross-modal integration is typically seamless, yet it remains unclear whether OLLMs demonstrate comparable consistency. When the same semantic content is presented in different forms—spoken audio, written text, or visual images—do models still converge on the same correct answer? We refer to this property as *cross-modal consistency*: the ability to maintain stable predictions regardless of input modality, thereby demonstrating reasoning over shared semantic representations rather than relying on modality-specific cues. Although directly diagnosing whether current OLLMs achieve this goal is non-trivial, we can evaluate them through carefully designed benchmarks that expose inconsistencies. For instance, by posing semantically identical questions under different modality settings, we can test whether predictions diverge across modalities — an indicator of reliance on surface-level patterns rather than genuine modality-invariant reasoning.

Recent benchmarks have taken promising steps toward evaluating OLLMs, particularly through audio-visual tasks that reveal baseline cross-modality performance. Datasets such as Music AVQA (Li et al., 2022), AV-Reasoner (Lu et al., 2025), and Pano-AVQA (Yun et al., 2021) primarily probe fine-grained audio–visual reasoning, while broader efforts like AVQA (Yang et al., 2022), WorldSense (Hong et al., 2025), AV-Odyssey Bench (Gong et al., 2024), and OmniBench (Li et al., 2024b) expand to general multimodal understanding across diverse contexts. However, these benchmarks largely overlook whether models remain consistent across modalities. While other works (Park et al., 2025; Zhang et al., 2024) attempt to assess modality consistency, they are restricted to the vision–text setting within vision–language models.

To address this gap, we introduce **XModBench**, a benchmark specifically designed to evaluate cross-modal consistency in omni-modal large language models. We formulate all questions in a multiple-choice format, where each question naturally contains two components: (i) a *context* describing an object or event, and (ii) a set of *candidates* from which the model must select the correct one. Unlike prior benchmarks that typically fix either the context or the choices to a single modality (Yang et al., 2022; Li et al., 2024b), XModBench systematically covers all six cross-modal directions among audio, vision, and text (see Tab. 1). To ensure broad coverage and rigorous evaluation, XMODBENCH spans five domains—perception, spatial reasoning, temporal reasoning, linguistic understanding, and external knowledge. We curate data across these domains through re-annotation, synthetic construction, and targeted web collection, ensuring both diversity and balance across modalities. The resulting benchmark comprises **61,320** multiple-choice question–answer pairs (10,220 unique instances), each instantiated in six modality configurations that preserve identical semantics across audio, visual, and textual forms. This enables both large-scale evaluation and fine-grained diagnosis of cross-modal consistency. An overview of the benchmark design is illustrated in Fig. 1.

Table 1: Comparison of multimodal question-answering (QA) benchmarks by modality coverage, task domains, and modality consistency.

| Benchmark | #Q | Context Modality | | | Candidate Modality | | | Task Domain | | | | | Mod. Consist. |
|---|---|---|---|---|---|---|---|---|---|---|---|---|---|
| | | Text | Vision | Audio | Text | Vision | Audio | Percep. | Spatial | Temporal | Ling. | Ext. Know. | |
| MME Bench (Fu et al., 2024a) | 2,194 | ✗ | ✓ | ✗ | ✓ | ✗ | ✗ | ✓ | ✗ | ✗ | ✓ | ✓ | ✗ |
| MMBench (Liu et al., 2024) | 3,217 | ✗ | ✓ | ✗ | ✓ | ✗ | ✗ | ✓ | ✓ | ✗ | ✓ | ✓ | ✗ |
| OcrBench v2 (Fu et al., 2024b) | 10,000 | ✗ | ✓ | ✗ | ✓ | ✗ | ✗ | ✓ | ✗ | ✗ | ✗ | ✗ | ✗ |
| SEED-Bench-2 (Li et al., 2024a) | 24,371 | ✓ | ✓ | ✗ | ✓ | ✓ | ✗ | ✓ | ✓ | ✓ | ✓ | ✓ | ✗ |
| AudioBench Wang et al. (2024) | 24,371 | ✗ | ✗ | ✓ | ✓ | ✗ | ✗ | ✓ | ✗ | ✗ | ✓ | ✗ | ✗ |
| Audiopedia (Li et al., 2022) | 45,867 | ✗ | ✗ | ✓ | ✓ | ✗ | ✗ | ✗ | ✗ | ✗ | ✓ | ✓ | ✗ |
| MMAU (Sakshi et al., 2024) | 10,000 | ✗ | ✗ | ✓ | ✓ | ✗ | ✗ | ✓ | ✗ | ✗ | ✓ | ✗ | ✗ |
| AVQA (Yang et al., 2022) | 57,335 | ✗ | ✓ | ✓ | ✓ | ✗ | ✗ | ✓ | ✓ | ✓ | ✗ | ✗ | ✗ |
| Pano-AVQA (Yun et al., 2021) | 51,700 | ✗ | ✓ | ✓ | ✓ | ✗ | ✗ | ✓ | ✓ | ✗ | ✗ | ✗ | ✗ |
| Music-AVQA (Li et al., 2022) | 45,867 | ✗ | ✓ | ✓ | ✓ | ✗ | ✗ | ✓ | ✓ | ✗ | ✗ | ✗ | ✗ |
| SAVE Bench (Sun et al., 2024) | 4,350 | ✗ | ✓ | ✓ | ✓ | ✗ | ✗ | ✓ | ✗ | ✗ | ✗ | ✗ | ✗ |
| Video-MME (Fu et al., 2025a) | 2,700 | ✗ | ✓ | ✓ | ✓ | ✗ | ✗ | ✓ | ✓ | ✓ | ✓ | ✓ | ✗ |
| WorldSense (Hong et al., 2025) | 3,172 | ✗ | ✓ | ✓ | ✓ | ✗ | ✗ | ✓ | ✗ | ✓ | ✗ | ✗ | ✗ |
| AV-Reasoner (Lu et al., 2025) | 1,027 | ✗ | ✓ | ✓ | ✓ | ✗ | ✗ | ✗ | ✗ | ✓ | ✗ | ✗ | ✗ |
| AV-Odyssey Bench (Gong et al., 2024) | 1,142 | ✗ | ✓ | ✓ | ✓ | ✓ | ✓ | ✓ | ✓ | ✗ | ✗ | ✓ | ✗ |
| OmniBench (Li et al., 2024b) | 4,555 | ✗ | ✓ | ✓ | ✓ | ✗ | ✗ | ✓ | ✗ | ✗ | ✓ | ✗ | ✗ |
| **XModBench (Ours)** | 61,320 | ✓ | ✓ | ✓ | ✓ | ✓ | ✓ | ✓ | ✓ | ✓ | ✓ | ✓ | ✓ |

We systematically evaluate models on XMODBENCH, going beyond overall accuracy to provide fine-grained diagnosis of cross-modal reasoning. Specifically, we analyze three complementary dimensions: (1) **Task competence**—by averaging over all six modality directions, we assess model performance across perception, spatial, temporal, linguistic, and knowledge tasks, yielding task-centric comparisons of multimodal competence; (2) **Modality disparity**—we measure consistency when the same question is posed in different modalities, where high variability signals reliance on modality-specific cues rather than shared semantic representations; and (3) **Directional imbalance**—we compare accuracy when context and candidate modalities are swapped, revealing asymmetries in cross-modal grounding.

Our experiments show that current OLLMs fall short along all three axes. They perform strongly on perception and linguistic tasks (best models reach around 70%), but degrade by 15–25 points on spatial and temporal reasoning. Performance also drops sharply whenever audio is involved, underscoring that auditory representations remain the weakest link. Finally, accuracy is consistently higher when text serves as the candidate modality, highlighting incomplete bidirectional alignment across modalities. Together, these findings demonstrate that today's OLLMs remain far from achieving modality-invariant reasoning, underscoring the diagnostic value of XMODBENCH.

In summary, XMODBENCH makes the following key contributions:

1. **Cross-modal consistency benchmark.** We present XMODBENCH, the first tri-modal multiple-choice question-answering benchmark explicitly designed to evaluate cross-modal consistency, covering all six modality mappings among audio, vision, and text.

2. **Comprehensive coverage.** The benchmark spans five task families with 17 subtasks and 61,320 question–answer pairs, ensuring broad domain coverage and fine-grained diagnostics, while its balanced design enables fair assessment of modality-invariant reasoning.

3. **Diagnostic metrics.** We introduce *modality disparity* and *directional imbalance* to directly measure robustness and bidirectional alignment across modalities. Our experiments reveal systematic weaknesses in current OLLMs, providing actionable insights for developing more modality-invariant architectures and training strategies.

## 2 RELATED WORK

**Multimodal Question Answering (QA) Benchmarks.** A number of benchmarks have been developed to evaluate multimodal large language models (MLLMs). Grouped by modality composition, Yin et al. (2024), Liu et al. (2024), and Li et al. (2024a) focus on the vision–text setting (covering both images and videos). For audio–text evaluation, representative efforts include Wang et al. (2024) and Sakshi et al. (2024). When combining audio and vision with text, a variety of benchmarks have emerged, including Yang et al. (2022), Li et al. (2022), Yun et al. (2021), Sun et al. (2024), Hong et al. (2025), Lu et al. (2025), Gong et al. (2024), Li et al. (2024b), and Zhou et al. (2025). Other recent works, such as Yang et al. (2025), further extend evaluation to diverse multimodal combinations. Despite their breadth, these benchmarks primarily emphasize coverage across tasks and modalities, while less attention has been paid to evaluating *cross-modal consistency*—whether models produce stable answers when the same semantic content is expressed in different modality forms. Our work fills this gap by explicitly designing a benchmark centered on modality-invariant reasoning.

**Cross-Modality Consistency.** Recent work has begun to investigate whether multimodal models behave consistently across modalities. Park et al. (2025) introduced the Modality Importance Score to quantify modality bias, which measures how much each modality contributes to answering questions in VideoQA. Zhang et al. (2024) further proposed the notion of cross-modal consistency between text and image, defining a consistent model as one that applies the same internal reasoning to semantically identical inputs across modalities, thereby yielding consistent outcomes. In contrast, other studies, such as Sung-Bin et al. (2024) and Choong et al. (2024), report instances of inconsistent audio-video reasoning, where models hallucinate non-existent sounds or visual signals, thereby exposing modality bias and cross-modal inconsistency. While these efforts provide pioneering insights into cross-modal consistency, they are typically confined to specific modality pairs. Our work not only expands the scope to cover a broader range of modality combinations for state-of-the-art OLLMs, but also conducts a deeper analysis of their cross-modality reasoning behavior on a comprehensive task suite.

## 3 XModBench: Comprehensive cross-modal balanced benchmark

We introduce **XModBench**, a comprehensive multiple-choice question-answering (QA) benchmark designed to evaluate the cross-modal capabilities and consistency of OLLMs across audio, vision, and text. A key feature of **XModBench** is its modality-balanced design, which creates six cross-modal variants of semantically identical questions to enable a controlled and fair evaluation of cross-modal capabilities and consistency (Sec. 3.1). The benchmark offers extensive domain coverage through five task families and seventeen subtasks (Sec. 3.2), all built upon meticulously curated, high-quality, and diverse tri-modal data (Sec. 3.3).

### 3.1 Benchmark Design

The central objective of XModBench is to evaluate whether models preserve *cross-modal consistency* when the same semantic content appears in different modalities. Each item is a four-choice multiple-choice question consisting of a `<context>` (question stem) and four `<candidates>` (answer options). By systematically permuting text (T), vision (V), and audio (A) across the `<context>` and `<candidates>`, we generate six modality configurations of the same question (see Fig. 1 (b) and (d)). This balanced design ensures that no single modality is privileged and enables consistent evaluation across all directions, which supports three diagnostic properties aligned with the goals of our benchmark:

**(1) Task competence.** Since each task is instantiated uniformly across all modality pairs, we measure competence by averaging accuracy across all context–candidate configurations. This yields a fair estimate of a model's overall capability for each task, independent of modality-specific biases.

**(2) Modality disparity.** By presenting semantically identical questions under different modality configurations, we keep the content fixed while varying only the modality. For example, to compare audio and vision, we examine cases where text provides the context with audio candidates (T↦A) versus text with visual candidates (T↦V), and similarly compare A↦T against V↦T settings. Differences in accuracy under these controlled comparisons reveal modality disparities, indicating the relative competence across different modalities.

**(3) Directional imbalance.** We examine inverse settings by swapping the modalities of context and candidates. For example, a model may perform well when vision serves as the context and text provides the options (V↦T), but perform worse when the same semantic content is presented as a text context with visual candidates (T↦V). Such differences indicate asymmetric grounding between the two modalities, and comparable asymmetries are also observed in the audio–text and audio–vision pair.

### 3.2 Task Taxonomy

XModBench covers five task families with seventeen subtasks, spanning perception, spatial reasoning, temporal reasoning, linguistic understanding, and external knowledge (see Fig. 2). Each task is formulated in the multiple-choice format and follows the modality-balanced configuration described in Section 3.1: a `<context>` is drawn from one modality and four `<candidates>` from another. In this section, we detail the design of these subtasks and specify how each instance is instantiated across modalities within every task.

**Task 1. Perception.** This task evaluates whether models can recognize the same object, activity, or scene across modalities. For example, a barking dog may appear as an image, as its sound, or as the text description "dog barking." Here, visual inputs are images, audio inputs are recordings of corresponding sounds, and text inputs are short labels or phrases. The data are drawn from diverse domains, including human activities, animal behaviors, musical instruments, and natural environments.

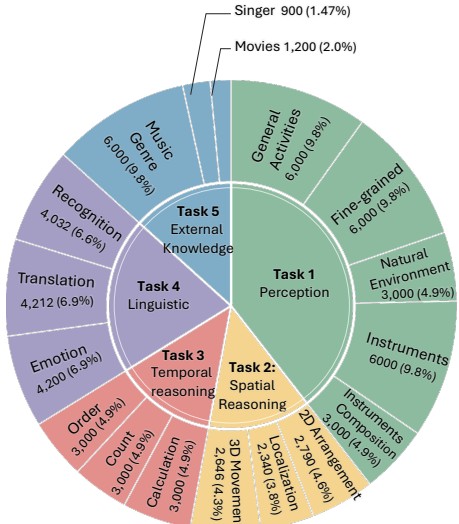

Figure 2: Distribution of XModBench's questions across five task families with specific subtasks.

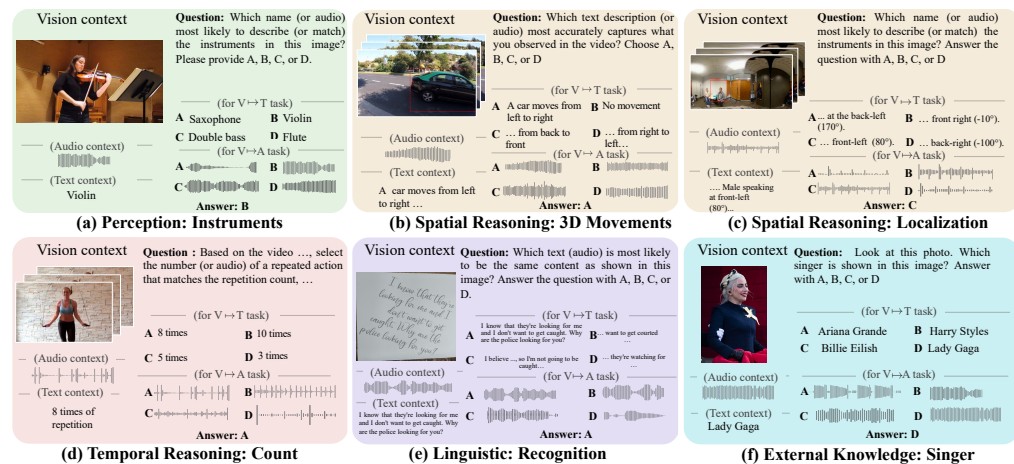

Figure 3: XModBench task examples. We show sample questions from six subtasks in the benchmark. Each question includes possible contexts from different modalities, and for the vision-context example, the candidates are given in either text or audio.

We divide perception into several subtasks. **General activity** recognition mixes candidates from diverse domains to test broad semantic alignment, while **fine-grained activity** recognition restricts candidates to a single domain (e.g., animal species or instrument types), thereby increasing difficulty and requiring precise discrimination. We further design domain-specific subtasks to capture unique challenges: recognizing **natural environments** (e.g., rainfall, wind, fire), distinguishing **instruments** (e.g., violin, bass, cello), and identifying **instrument compositions** where multiple instruments are played together (e.g., violin and bass, or cello and flute). Illustrative examples are shown in Fig. 1(d) and Fig. 3(a).

**Task 2. Spatial reasoning.** This task evaluates whether models can interpret object positions and motion in 2D and 3D space, which is an important factor in vision–language models (Chen et al., 2024). We extend this ability to the omni-modal setting and design three subtasks. The first is **2D arrangement**, where the model determines the left–right order of objects such as musical instruments. Visual inputs are images of ordered layouts, audio inputs are stereo recordings with left–right cues, and text inputs describe the relative arrangement; distractors are generated by swapping or permuting positions. The second subtask, **3D localization**, using panoramic videos from Shimada et al. (2023), requires identifying the orientation of events in video frames, spatialized audio, and short textual descriptions (e.g., "a man speaking from the front-left"); distractors are produced by shifting the same scene to nearby but incorrect directions through camera or audio rotation. The third subtask, **3D movement understanding**, focuses on motion directions such as left–right or front–back, instantiated with street-view or action videos, spatialized audio of approaching or receding sounds, and textual trajectory descriptions (Fuentes et al., 2022); distractors are clips with incorrect motion patterns or mismatched vehicle types. Examples for the 3D movement and localization tasks are shown in Fig. 3(b) and (c), respectively.

**Task 3. Temporal reasoning.** This task evaluates whether models can understand **event order** and **frequency** across time in video and audio. We design three subtasks. The first is **temporal order**, where models infer the correct sequence of events from muted video segments, audio clips, or textual descriptions and align them across modalities. The second, **temporal counting**, requires recognizing the number of repeated actions such as tennis hits, jumps, or drum beats, with distractors differing in count. For example, a video may show a tennis player hitting the ball three times, and the model must select the audio clip with exactly three hits or the text "3 times." The third, **temporal calculation**, extends counting by applying simple arithmetic to the repetition number. For instance, if a video shows a person jumping three times and the query applies $2 \times$ count, the correct answer should correspond to six repetitions, given either as an audio clip with six jumps or as the text "6 times." An example of the temporal counting task is in Fig. 3(d).

**Task 4. Linguistic understanding.** This task covers recognition of linguistic content and interpretation of affective meaning. While prior work separates OCR for vision and ASR for audio (Fu et al., 2024b; Wang et al., 2024), XModBench unifies them in a cross-modal setting. We design

three subtasks. The first, **linguistic recognition**, focuses on transcribing text from images, audio, or phrases; correct candidates require word-level precision, while distractors differ by only one or two words (see Fig. 3(e)). The second, **translation**, evaluates English–Chinese translation across modalities, with distractors introducing subtle shifts such as antonyms, degree modifiers (e.g., "very" → "a little"), or small changes in numbers and entities. The third, **emotion classification**, targets affective understanding in dialogue: audio inputs are spoken conversations, visual inputs are muted video clips with transcripts, and candidates represent emotions such as joy, sadness, or anger, with distractors drawn from closely related categories.

**Task 5. External knowledge.** Beyond perceptual and reasoning skills, some tasks require linking multimodal content with world knowledge. We design three subtasks. The first, **movie recognition**, presents audio clips from trailers, visual posters, or short text descriptions of the plot, with candidates drawn from films of similar genres or storylines. The second, **music genre classification**, uses album covers, short audio clips, or textual genre labels, with distractors from closely related genres (e.g., "jazz" vs. "blues"). The third, **singer identification**, provides names, portrait images, or audio clips of songs, with distractors sampled from artists of similar musical styles (see Fig. 3(f)).

### 3.3 Data Curation

The construction of XModBench follows a three-stage pipeline. We begin by collecting large-scale text–vision–audio triplets across all task domains, then generate task-specific multiple-choice questions, and finally apply both automated filtering and human verification to ensure quality and consistency.

**Cross-modal data collection.** We curate a large corpus aligned across vision, audio, and text by combining three sources: (i) re-annotated and extended data from existing multimodal datasets, such as adapting VGG-Sound for perception tasks or STARSS23 (Lee et al., 2022; Shimada et al., 2023) for spatial reasoning; (ii) synthetic or model-generated content to cover missing modalities, for example generating speech audio with FireRedTTS (Guo et al., 2025) or producing rendered text images for translation tasks; and (iii) web-collected samples for domains not well represented in public resources, such as singer portraits and songs for the *Singer Identification* task or trailers and posters for *Movie Recognition* from public YouTube videos. This design ensures both coverage and balance across all five task families. Detailed sources and processing procedures are described in Appendix G.

**Question candidate generation.** To ensure the correctness of both the generated questions and answers, we first construct task-specific multiple-choice templates using our annotated tri-modality data. The question descriptions are then refined by GPT-5 (OpenAI, 2025) solely to improve language fluency and stylistic diversity. Importantly, this refinement does not introduce any new information or alter the underlying semantics. Each question is instantiated with a context and four candidates under the modality-balanced configuration, ensuring consistent evaluation across all modality directions. Distractors are created to be semantically challenging but unambiguous, while templates are diversified with both human-written prompts and LLM-assisted variations.

**LLM filtering and human-in-the-loop verification.** To guarantee data quality at scale, we first adopt foundation models (OpenAI, 2025; Comanici et al., 2025) to filter out low-quality or ambiguous samples. Human annotators then double-check the filtered results to ensure accuracy. After questions are constructed, an internal round of testing is conducted by annotators, who resolve ambiguities and validate correctness. Feedback from this process is used to regenerate and retest questions until high-quality items are obtained.

Overall, this pipeline yields a high-quality benchmark with diverse and well-aligned multimodal content. More detailed descriptions of dataset sources, generation strategies, and signal-processing techniques are provided in Appendix G.

## 4 Experiments

### 4.1 Baselines

We evaluate XModBench on a diverse set of recent omni-modal large language models. The **Gemini series** (Comanici et al., 2025; Team et al., 2024) represents state-of-the-art closed-source omni-modal models, and we include multiple variants ranging from Gemini 1.5 Pro to Gemini 2.5 Pro. Note that OpenAI APIs do not currently support processing audio and visual modalities jointly

Table 2: Results on **XModBench**. We report (a) the performance under different input modalities across the full benchmark, and (b) the summary of average accuracy for each of the 5 task families. The highest scores are **bolded**, and the second highest are underlined.

| Model | Accuracy on 5 Task Families | | | | | Modality Configuration | | | | | | | Avg. |
|---|---|---|---|---|---|---|---|---|---|---|---|---|---|
| | Perc. | Spat. | Temp. | Ling. | Knwl. | $A \mapsto T$ | $A \mapsto V$ | $T \mapsto A$ | $T \mapsto V$ | $V \mapsto A$ | $V \mapsto T$ | *Std.* | |
| No Context | 25.5 | 24.8 | 24.9 | 24.7 | 25.5 | 25.1 | 24.3 | 25.4 | 24.8 | 25.3 | 25.7 | 0.4 | 25.1 |
| Qwen2.5-VL | 91.3 | 51.4 | 40.9 | 84.1 | 77.2 | - | - | - | 60.1 | - | 74.7 | - | 67.4 |
| Intern3.5-VL | 87.2 | 42.7 | 41.4 | 75.0 | 68.7 | - | - | - | 49.7 | - | 73.7 | - | 61.7 |
| PandaGPT | 24.6 | 25.7 | 24.4 | 25.5 | 23.1 | 24.5 | 25.0 | 23.8 | 25.2 | 24.5 | 25.1 | 0.5 | 24.7 |
| Unified-IO 2 | 36.1 | 23.6 | 23.8 | 30.4 | 26.8 | 28.9 | 24.0 | 25.4 | 32.0 | 25.7 | 32.7 | 3.7 | 28.1 |
| Unified-IO 2 XL | 42.2 | 25.0 | 26.1 | 30.8 | 29.5 | 33.3 | 27.0 | 27.1 | 32.9 | 26.5 | 37.4 | 4.5 | 30.7 |
| Unified-IO 2 XXL | 43.7 | 28.3 | 27.7 | 31.2 | 34.0 | 37.4 | 25.0 | 31.2 | 37.8 | 26.7 | 39.9 | 6.3 | 33.0 |
| VideoLLaMA 2 | 45.7 | 33.9 | 29.2 | 36.7 | 36.8 | 48.6 | 26.0 | 25.7 | 26.5 | 25.2 | 66.8 | 17.4 | 36.5 |
| VITA | 34.8 | 34.0 | 29.4 | 46.1 | 32.6 | 40.2 | 26.0 | 29.8 | 26.8 | 29.9 | 59.3 | 12.8 | 35.4 |
| Baichuan Omni 1.5 | 58.9 | 34.9 | 30.0 | 62.8 | 56.7 | 47.8 | 35.8 | 40.5 | 56.2 | 38.6 | 73.0 | 14.0 | 48.7 |
| EchoInk-R1 | 75.8 | 36.6 | 37.1 | 73.3 | 73.3 | 64.6 | 45.9 | 56.4 | 60.9 | 49.9 | 77.6 | 11.3 | 59.2 |
| Qwen2.5-Omni | 75.5 | 38.4 | 32.3 | 74.1 | 72.8 | 62.0 | 48.0 | 55.4 | 59.6 | 50.5 | 76.3 | 10.1 | 58.6 |
| Gemini 1.5 Pro | 56.2 | 40.1 | 37.1 | 72.6 | 69.4 | 52.4 | 38.2 | 48.6 | 70.4 | 40.7 | 79.9 | 16.7 | 55.0 |
| Gemini 2.0 Flash | 66.2 | 48.4 | 44.8 | 70.2 | 78.1 | 63.7 | 49.0 | 52.2 | 71.5 | 47.6 | 85.2 | 15.2 | 61.2 |
| Gemini 2.5 Flash | 66.1 | 48.0 | 48.6 | 73.1 | 82.8 | 62.6 | 51.2 | 55.1 | 75.7 | 51.9 | **86.0** | 14.2 | 63.7 |
| Gemini 2.5 Pro | **75.9** | **50.1** | **60.8** | **76.8** | **89.3** | **71.0** | **58.9** | **64.4** | **79.8** | **60.8** | 88.6 | 11.7 | **70.6** |
| Human | 91.0 | 89.7 | 88.9 | 93.9 | 93.9 | 92.4 | 91.5 | 91.1 | 91.8 | 86.4 | 95.6 | 3.0 | 91.5 |

within a single query; therefore, we omit the GPT series from our evaluation. For open-source models, we include the latest **Qwen2.5-Omni** (Xu et al., 2025), **Baichuan Omni 1.5** (Li et al., 2025), and **EchoInk-R1** (Xing et al., 2025). Additional open-source omni-modal baselines include **VideoLLaMA 2** (Cheng et al., 2024), **VITA** (Fu et al., 2025b), the **Unified-IO 2** series (Large, XL, and XXL variants) (Lu et al., 2024), and **PandaGPT** (Su et al., 2023). Together, these models represent a broad spectrum of both closed- and open-source OLLMs.

## 4.2 MODEL PERFORMANCES

Table 5 reports results across five task families and six cross-modal directions (Audio ↦ Text, Audio ↦ Vision, Text ↦ Audio, Text ↦ Vision, Vision ↦ Audio, Vision ↦ Text). The first subtable summarizes the average accuracy across all tasks for each modality configuration, while the remaining subtables present detailed performance within each task family. The highest scores are **bolded**, and the second highest are underlined. For each model, we also report the overall average accuracy (*Avg.*) and standard deviation (*Std.*) across the six configurations to quantify robustness to modality shifts. Details of the human evaluation are provided in Appendix F.

**Performance by task families.** Overall, the Gemini 2.0 and 2.5 series outperform all open-source models. Among open models, Qwen2.5-Omni and EchoInk-R1 are the strongest baselines, surpassing Gemini 1.5 Pro by 3.6 and 4.2 points, respectively. Across the five task families, spatial and temporal reasoning remain the most challenging (Gemini 2.5 Pro achieves 50.1 and 60.8), whereas perception and linguistic tasks reach higher accuracy (75.9 and 76.8). The performance gap between open- and closed-source models extends beyond spatial and temporal reasoning to external knowledge: while Qwen2.5-Omni and EchoInk-R1 perform comparably to Gemini 2.5 Pro on perception, the latter attains 89.3 on external knowledge. These results highlight persistent bottlenecks in open-source models, as closed-source models likely benefit from broader web-scale pretraining and stronger spatial–temporal reasoning capabilities.

**Performance by modality configurations.** We also analyze performance consistency across modality configurations on the same tasks and observe clear divergences. Vision–text settings consistently outperform audio–text ones, confirming that visual representations are more strongly grounded than audio. In perception tasks, accuracy exceeds 90% with vision–text inputs but drops by over 20 points with audio–text. Audio–vision combinations without textual anchors yield the lowest scores, highlighting the difficulty of aligning heterogeneous signals. Among SOTA models, Gemini 2.5 Pro (Avg. 70.6, Std. 11.7) shows the best balance of accuracy and stability, while Qwen2.5-Omni (Std. 10.1) and EchoInk-R1 (Std. 11.3) are the most consistent open models. By contrast, Gemini 1.5 Pro and Baichuan Omni 1.5 have standard deviations exceeding 14, reflecting weaker robustness to modality variation.

### 4.3 MODALITY DISPARITY ANALYSIS

A key challenge for OLLMs is whether they handle audio, vision, and text equally rather than favoring one modality. XMODBENCH enables this by instantiating identical semantics across modality settings. The disparity is defined as $\Delta_{T \text{ vs. } V} = (Acc_{A \mapsto V} - Acc_{A \mapsto T}) + (Acc_{V \mapsto A} - Acc_{T \mapsto A})$,

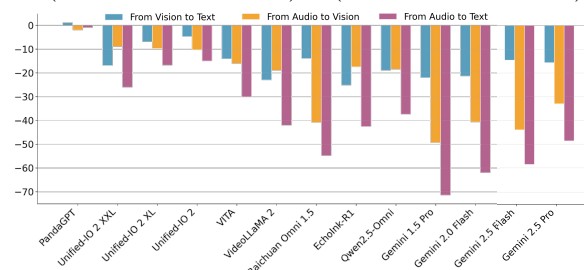

which quantify the paired subtraction, e.g., compares configurations that differ only by substituting **text** with **vision**, thereby isolating the effect of modality substitution on accuracy. Results in Fig. 4 show that $\Delta_{T \text{ vs. } A}$ exhibits the strongest disparity ($-49$ for Gemini 2.5 Pro), $\Delta_{V \text{ vs. } A}$ is moderate ($-33$), and $\Delta_{T \text{ vs. } V}$ remains smallest ($-15$). *These findings highlight audio as the most challenging modality, with vision showing moderate gaps and text remaining the most robust.*

Figure 4: Modality disparity across different configurations. Negative scores indicate performance gaps, with the largest disparities observed between audio and text.

### 4.4 DIRECTIONAL IMBALANCE

We test whether models behave consistently when swapping the roles of context and candidates. We define *directional imbalance* as $\Delta_{X \leftrightarrow Y} = Acc(X \mapsto Y) - Acc(Y \mapsto X)$, the accuracy gap between inverse configurations for $(X, Y) \in \{(A, T), (V, T), (V, A)\}$. As shown in Fig. 5, vision–text and audio–text pairs exhibit notable asymmetries: Gemini 2.5 Pro drops by $8.8$ points from T$\mapsto$V to V$\mapsto$T, and Qwen2.5-Omni shows a 16.6-point gap, while audio–text differences remain around 6–8 points. By contrast, **audio–vision** pairs are nearly symmetric but achieve much lower overall accuracy. *These findings suggest that directional imbalance mainly arises in text–vision and audio–text pairs, likely reflecting training data biases toward text as the dominant output modality.*

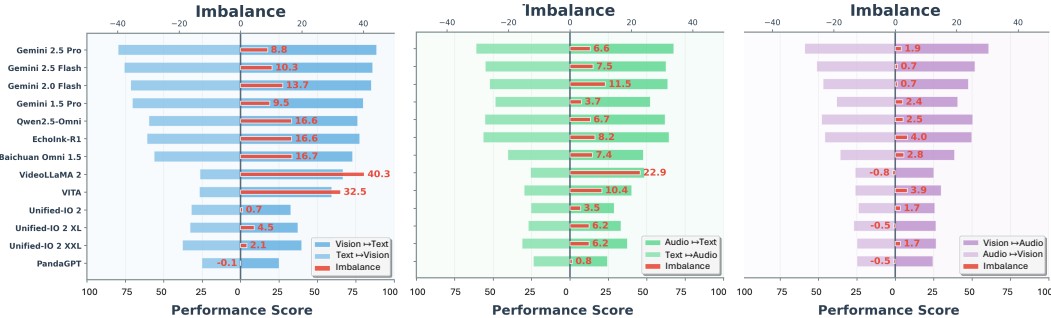

Figure 5: Directional imbalance: accuracy gaps between paired inverse settings among audio, vision and text. Models show clear asymmetries, especially in vision–text and audio–text pairs.

### 4.5 FAILURE CASE ANALYSIS

To better understand model errors, we prompt models like Gemini 2.5 Pro and Qwen2.5-Omni to generate reasoning alongside their answers. As shown in Fig. 6, we observe common failure cases that reflect modality performance gaps and alignment issues. Example (a) shows a mismatch between audio-to-text and audio-to-image reasoning: while the model correctly identifies a didgeridoo by text, it fails to select the matching image, revealing inconsistent grounding. In example (b), Qwen-2.5 Omni misinterprets spatial audio motion when switching from audio-to-text to text-to-audio, reversing the vehicle's direction. These errors highlight persistent asymmetries in cross-modal reasoning that only emerge when the same task is posed across different modality combinations.

### 4.6 TRIPLE-DOMAIN QUESTION ANSWERING

Real-world omni-modal scenarios often present information jointly across modalities rather than in isolation. To approximate this setting, we extend XMODBENCH tasks to an audio–visual context, where both sound and vision are provided in the question stem, while the candidates remain in text.

We evaluate this dual-context configuration using the Gemini series. Compared with single-modality baselines, the results (see Appendix E) show modest but consistent gains, indicating that models can

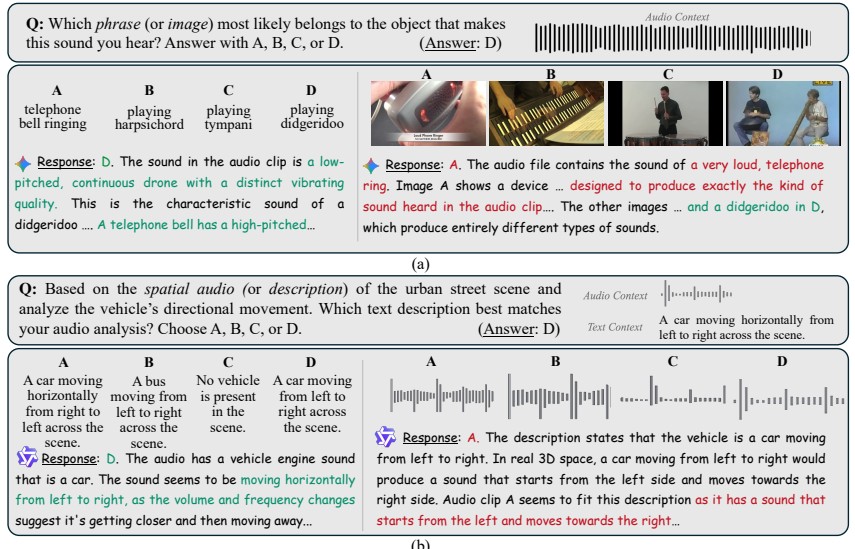

Figure 6: Failure cases. (a) Gemini 2.5 pro correctly identifies a didgeridoo in text but fails to match it with the corresponding image candidates. (b) shows Qwen2.5-Omni misinterprets spatial motion when switching candidates from text to audio. This cases illustrate asymmetries in cross-modal reasoning.

benefit from simultaneous multimodal cues. However, the improvements are not always additive, suggesting that current models do not yet fully exploit complementary signals across modalities.

Table 3: Performance of Gemini models using both audio and visual context $(A + V)$. Comparing $A + V$ with the best single-modality result $(\max(A, V))$ reveals whether adding more context leads to better reasoning or causes harmful interference.

| Setting | Gemini 1.5 Pro | Gemini 2.0 Flash | Gemini 2.5 Pro |
|---|---|---|---|
| A $\mapsto$ T | 52.76 | 63.71 | 70.99 |
| V $\mapsto$ T | 79.92 | 85.20 | 88.60 |
| A+V $\mapsto$ T | 82.53 (**+2.61**) | 79.84 (**-5.36**) | 89.76 (**+1.16**) |

## 5 KEY TAKEAWAYS FOR OLLM DEVELOPMENT

Our benchmark results serve as a diagnostic tool, revealing how underlying **data composition** and **training methodologies** shape OLLMs behaviors. By correlating performance patterns in 4.3 and 4.4 with model architectures and training reports, we derive three critical insights regarding *interleaved data*, *task coverage*, and how these effect OLLMs after *post-training*.

### 5.1 INTERLEAVED DATA REDUCES DIRECTIONAL IMBALANCE

A key observation from our imbalance analysis is the connection between interleaved training data and the reduction of directional imbalance when modalities are swapped, as seen in the results from Section 4.4.

Public official reports indicate that models such as *Qwen2.5-Omni* (Xu et al., 2025) and *Gemini-2.5* series (Comanici et al., 2025) incorporate massive scale interleaved multimodal corpora (e.g., mixed audio-vision conversations). Our benchmark corroborates this: these models exhibit relatively small performance gaps between Audio→Vision and Vision→Audio tasks. This suggests that encountering modalities interchangeably in context allows the model to build symmetric cross-modal bridges.

Conversely, models trained primarily on non-interleaved datasets exhibit significant directional asymmetry. For instance, in the case of VideoLLaMA (Cheng et al., 2024), despite having strong backbones, models relying on open-source data with limited interleaved audio-vision instruction pairs show a distinct bias. Models fail to generalize when the modalities are swapped (e.g., needing

to distinguish between four video or audio clips), indicating that insufficient interleaved supervision hinders directional robustness.

## 5.2 Task Coverage Gaps

With its broad domain coverage, **XModBench** serves as a diagnostic tool to reveal blind spots in training for both open- and closed-source models. It identifies these gaps through performance inconsistencies, particularly regarding the diversity of audio types.

- **Spoken vs. Non-Spoken Bias:** Many OLLMs prioritize speech as the primary modality for audio data. Models relying on pre-trained, speech encoders (e.g., Whisper (Radford et al., 2023) in the *Baichuan* (Li et al., 2025) model) and trained predominantly on spoken language tasks exhibit a sharp performance decline in non-spoken audio domains, such as environmental sound recognition and spatial reasoning. This indicates a significant semantic gap between linguistic speech and general acoustic signals, resulting in a weakness for comprehensive audio understanding.

- **Deficiencies in Music Understanding and Spatial Reasoning:** Distinct gaps in specific categories shows evidence of missing training data. For example, despite its high overall capacity, *Gemini 1.5* demonstrates limited capability in musical reasoning, suggesting an absence of related data in its training corpus. Similarly, *EchoInk-R1* struggles with spatial-vision tasks relative to related families, pointing to a lack of spatial-centric visual content.

## 5.3 The Impact of Post-Training on Alignment

A comparative analysis of *EchoInk-R1* and the *Qwen2.5-Omni* series illustrates how post-training strategies can reshape and even degrade multimodal alignment when data is weakness, despite the advance reinforcement learning used to enhances cross-modal reasoning (Shao et al., 2024). *Qwen2.5-Omni* integrates interleaved multimodal conversations during post-training, *EchoInk-R1* is further fine-tuned on the data derived from OmniInstruct corpus (Li et al., 2024b) focus on spoken instruction following. This divergence in training objectives leads to notable behavioral differences:

- Although *EchoInk-R1* is based on the well-aligned *Qwen2.5-Omni*, it shows **larger modality disparities and directional imbalances**. Fig.4 shows it performs worse in Audio→Text and Vision→Text modality disparity, and exhibits higher imbalance in Audio-Vision and Audio-Text pairs in Fig. 5. This suggests that the cross-modal alignment from interleaved pre-training can be eroded if the post-training stage shifts to non-interleaved data.

- Despite heavy tuning, *EchoInk* fails to improve in spatial tasks (Fig. 2), showing that post-training requires **comprehensive task coverage**. Incomplete fine-tuning cannot fill these gaps and may even degrade the model's multimodal capabilities.

Beyond simple evaluation, XModBench serves as a diagnostic tool for model development. While training data for SOTA models remains a black box, XModBench reveals the impact of training strategies and task coverage through controlled comparisons from them. Our open-source generation tools also offer the potential to extend to new domains. By using a unique modality-swap design, XModBench exposes hidden flaws that current benchmarks miss, making it essential for building robust next generation omni-modal language models.

## 6 Conclusion

We introduced **XModBench**, a benchmark for diagnosing cross-modal consistency in omni-language models. By systematically interleaving audio, vision, and text across diverse tasks, XMod-Bench enables fine-grained evaluation of modality disparities, directional imbalances, and modality invariant capability. Our results show that audio remains the most challenging modality, that models often behave asymmetrically in inverse settings such as text-vision and audio-text, and that combining audio and vision yields only modest gains. Overall, while current systems are strong in perception and language, they still lack stable and consistent reasoning across modalities, leaving ample room for progress toward truly modality-agnostic intelligence.

ETHICS STATEMENT

Our study does not involve private or sensitive personal data. All audiovisual samples are obtained from publicly available official sources, including previously published research datasets, content hosted on established open source platforms such as Hugging Face and Kaggle.

For all newly generated labels and annotations, we perform manual verification to ensure correctness and to remove any potentially inappropriate content. All web-curated data are from publicly accessible and previously published sources without requiring special authentication. All materials are used solely for non-commercial academic research. We do not redistribute copyrighted video or audio; only derived features, annotations, and evaluation results are released.

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
