## SUPPLEMENTARY MATERIAL

## TABLE OF CONTENT

# A  MINI BENCHMARK RESULT

We will release a standardized 6k-sample XModBench-Lite benchmark, consisting of 5 task families × 6 modality–configuration settings, with 200 examples per setting. The dataset is balanced across both task families and modality directions. The overall performance (see Tab. 4) trends and error patterns closely mirror those reported in Tab.2 of the main paper.

Table 4: Performance on 6k version of XModBench

| Model | Accuracy on 5 Task Families | | | | | Modality Configuration | | | | | | Avg. |
|---|---|---|---|---|---|---|---|---|---|---|---|---|
| | Perc. | Spat. | Temp. | Ling. | Knwl. | A → T | A → V | T → A | T → V | V → A | V → T | |
| w/o context | 25.3 | 25.1 | 24.8 | 24.4 | 25.2 | 26.5 | 24.8 | 24.2 | 24.1 | 25.5 | 25.1 | 25.0 |
| Qwen2.5-VL | 91.5 | 51.9 | 40.5 | 84.3 | 76.5 | - | - | - | 68.2 | - | 72.8 | 60.5 |
| Intern3.5-VL | 88.2 | 41.8 | 48.5 | 75.8 | 62.4 | - | - | - | 46.5 | - | 73.1 | 69.8 |
| PandaGPT | 24.9 | 25.3 | 23.8 | 24.7 | 21.3 | 25.2 | 25.5 | 22.8 | 24.9 | 24.8 | 23.1 | 24.4 |
| Unified-IO 2 | 36.5 | 24.8 | 24.1 | 31.2 | 27.5 | 29.8 | 24.2 | 25.5 | 32.1 | 25.9 | 33.5 | 28.4 |
| Unified-IO 2 XL | 42.5 | 26.3 | 26.0 | 32.5 | 30.6 | 33.8 | 26.8 | 27.5 | 34.2 | 27.1 | 38.8 | 31.2 |
| Unified-IO 2 XXL | 44.1 | 29.0 | 27.3 | 32.9 | 34.7 | 38.0 | 26.3 | 31.8 | 38.5 | 27.3 | 40.5 | 33.6 |
| VideoLLaMA 2 | 46.1 | 34.2 | 29.0 | 37.5 | 37.4 | 49.1 | 26.2 | 26.0 | 27.3 | 25.5 | 67.8 | 36.9 |
| VITA | 35.6 | 31.8 | 29.2 | 46.1 | 30.5 | 45.2 | 26.5 | 26.8 | 26.1 | 24.5 | 52.5 | 33.6 |
| Baichuan Omni 1.5 | 59.5 | 35.5 | 30.8 | 63.9 | 57.2 | 48.5 | 36.2 | 41.1 | 57.2 | 39.1 | 74.5 | 49.5 |
| EchoInk-R1 | 73.1 | 35.8 | 36.4 | 73.3 | 72.4 | 66.5 | 42.1 | 56.0 | 65.5 | 46.8 | 72.3 | 53.2 |
| Qwen2.5-Omni | **78.6** | 37.1 | 31.2 | 74.2 | 77.8 | 69.5 | 45.2 | 54.5 | 58.1 | 56.8 | 74.5 | 51.4 |
| Gemini 1.5 Pro | 56.8 | 40.8 | 38.0 | 71.0 | 69.9 | 53.1 | 38.5 | 49.2 | 71.2 | 41.4 | 80.4 | 55.7 |
| Gemini 2.0 Flash | 65.4 | 48.9 | 41.5 | 72.2 | 71.2 | 68.5 | 44.2 | 50.1 | 74.3 | 47.5 | **88.9** | 67.2 |
| Gemini 2.5 Flash | 66.5 | 47.1 | 45.3 | 74.4 | 81.2 | 64.8 | 49.4 | 57.5 | **77.2** | 51.0 | 81.6 | 68.6 |
| Gemini 2.5 Pro | 74.8 | **59.3** | **60.2** | **75.8** | **89.1** | **76.8** | **50.5** | **63.2** | 75.1 | **61.2** | 82.5 | **71.8** |

# B MODALITY CONFIGURATION SCORE UNDER FIVE TASK

Table 5 reports the detailed results for all six modality–configuration settings (A↦T, A↦V, T↦A, T↦V, V↦A, V↦T) across the five task families in XModBench (Perception, Spatial, Temporal, Linguistic, and Knowledge), as well as the overall average score on the full benchmark.

Table 5: Results on **XModBench** across 5 task families and 6 predefined cross-modal directions among **T**ext, **V**ision, and **A**udio. The first block reports the average accuracy across all tasks, followed by Task 1–5 (Perception, Spatial, Temporal, Linguistic, External knowledge). Scores are color-coded as < 30 , 30–60 , 60–90 , ≥ 90 , with the best in each column highlighted in **bold**.

**Overall Average**

| Model | A↦T | A↦V | T↦A | T↦V | V↦A | V↦T | Avg. | Std. |
|---|---|---|---|---|---|---|---|---|
| PandaGPT | 24.5 | 25.0 | 23.8 | 25.2 | 24.5 | 25.1 | 24.7 | 0.5 |
| Unified-IO 2 | 28.9 | 24.0 | 25.4 | 32.0 | 25.7 | 32.7 | 28.1 | 3.7 |
| Unified-IO 2 XL | 33.3 | 27.0 | 27.1 | 32.9 | 26.5 | 37.4 | 30.7 | 4.5 |
| Unified-IO 2 XXL | 37.4 | 25.0 | 31.2 | 37.8 | 26.7 | 39.9 | 33.0 | 6.3 |
| VideoLLaMA 2 | 48.6 | 26.0 | 25.7 | 25.2 | 66.8 | 36.5 | 17.4 | |
| VITA | 40.2 | 26.0 | 29.8 | 26.8 | 29.9 | 59.3 | 35.4 | 12.8 |
| Baichuan Omni 1.5 | 47.8 | 35.8 | 40.5 | 56.2 | 38.6 | 73.0 | 48.7 | 14.0 |
| EchoInk-R1 | 64.6 | 45.9 | 56.4 | 60.9 | 49.9 | 77.6 | 59.2 | 11.3 |
| Qwen2.5-Omni | 62.0 | 48.0 | 55.4 | 59.6 | 50.5 | 76.3 | 58.6 | 10.1 |
| Gemini 1.5 Pro | 52.4 | 38.2 | 48.6 | 70.4 | 40.7 | 79.9 | 55.0 | 16.7 |
| Gemini 2.0 Flash | 63.7 | 49.0 | 52.2 | 71.5 | 47.6 | 85.2 | 61.2 | 15.2 |
| Gemini 2.5 Flash | 62.6 | 51.2 | 55.1 | 75.7 | 51.9 | 86.0 | 63.7 | 14.2 |
| Gemini 2.5 Pro | **71.0** | **58.9** | **64.4** | **79.8** | **60.8** | **88.6** | **70.6** | 11.7 |
| Human | 92.4 | 91.5 | 91.1 | 91.8 | 86.4 | 95.6 | 91.5 | 3.0 |

**Task 1 - Perception**

| Model | A↦T | A↦V | T↦A | T↦V | V↦A | V↦T | Avg. | Std. |
|---|---|---|---|---|---|---|---|---|
| PandaGPT | 24.5 | 24.7 | 24.8 | 24.5 | 24.6 | 24.7 | 24.6 | 0.1 |
| Unified-IO 2 | 35.5 | 25.3 | 26.3 | 55.9 | 29.1 | 44.7 | 36.1 | 12.1 |
| Unified-IO 2 XL | 53.3 | 27.9 | 30.3 | 59.1 | 27.6 | 55.0 | 42.2 | 15.0 |
| Unified-IO 2 XXL | 55.0 | 26.9 | 39.0 | 64.2 | 26.7 | 50.2 | 43.7 | 15.4 |
| VideoLLaMA 2 | **74.7** | 26.6 | 28.3 | 26.8 | 26.5 | 91.5 | 45.7 | 29.4 |
| VITA | 37.1 | 25.4 | 27.0 | 23.7 | 26.4 | 69.1 | 34.8 | 17.5 |
| Baichuan Omni 1.5 | 42.7 | 36.3 | 45.6 | 87.8 | 50.3 | 90.7 | 58.9 | 24.0 |
| EchoInk-R1 | 74.1 | 58.5 | **69.3** | 91.6 | 67.7 | 93.4 | 75.8 | 13.9 |
| Qwen2.5-Omni | 72.9 | 59.1 | 69.2 | 91.2 | 68.5 | 92.0 | 75.5 | 13.3 |
| Gemini 1.5 Pro | 52.4 | 27.9 | 45.0 | 95.8 | 32.1 | 95.3 | 56.2 | 31.1 |
| Gemini 2.0 Flash | 56.8 | 45.0 | 54.2 | 92.7 | 55.1 | 93.4 | 66.2 | 21.2 |
| Gemini 2.5 Flash | 52.6 | 44.3 | 53.4 | 95.4 | 56.0 | 95.0 | 66.1 | 22.8 |
| Gemini 2.5 Pro | 62.3 | 57.4 | 64.4 | **97.1** | **72.6** | **97.6** | **75.9** | 17.4 |
| Human | 92.9 | 94.2 | 91.3 | 89.2 | 85.4 | 92.9 | 91.0 | 3.2 |

**Task 2 - Spatial**

| Model | A↦T | A↦V | T↦A | T↦V | V↦A | V↦T | Avg. | Std. |
|---|---|---|---|---|---|---|---|---|
| PandaGPT | 25.5 | 26.6 | 26.0 | 27.2 | 25.8 | 23.1 | 25.7 | 1.4 |
| Unified-IO 2 | 26.0 | 20.7 | 22.4 | 25.0 | 23.1 | 24.7 | 23.6 | 1.9 |
| Unified-IO 2 XL | 24.8 | 23.0 | 25.8 | 26.0 | 26.0 | 24.5 | 25.0 | 1.2 |
| Unified-IO 2 XXL | 29.6 | 23.6 | 30.9 | 25.5 | 29.5 | 30.7 | 28.3 | 3.0 |
| VideoLLaMA 2 | 43.9 | 27.8 | 24.4 | 27.5 | 25.2 | 54.3 | 33.9 | 12.3 |
| VITA | 42.3 | 28.9 | 24.6 | 30.9 | 25.1 | 52.2 | 34.0 | 11.0 |
| Baichuan Omni 1.5 | 38.1 | 28.0 | 25.1 | 31.7 | 25.3 | 61.2 | 34.9 | 13.8 |
| EchoInk-R1 | 41.3 | 27.2 | 26.8 | 34.0 | 28.0 | 62.2 | 36.6 | 13.7 |
| Qwen2.5-Omni | 41.8 | 31.2 | 26.7 | 34.4 | 28.6 | 67.8 | 38.4 | 15.3 |
| Gemini 1.5 Pro | 37.2 | 31.2 | 24.5 | 51.4 | 23.7 | 72.8 | 40.1 | 19.0 |
| Gemini 2.0 Flash | 45.2 | **43.1** | 29.2 | 56.4 | 33.5 | 83.0 | 48.4 | 20.4 |
| Gemini 2.5 Flash | **45.6** | 31.4 | 30.2 | 71.2 | 26.7 | 83.2 | 48.0 | 23.8 |
| Gemini 2.5 Pro | 41.0 | 32.9 | **32.1** | **75.8** | 30.3 | **88.3** | **50.1** | 25.4 |
| Human | 93.3 | 93.3 | 81.7 | 86.7 | 86.7 | 96.7 | 89.7 | 5.7 |

**Task 3 - Temporal**

| Model | A↦T | A↦V | T↦A | T↦V | V↦A | V↦T | Avg. | Std. |
|---|---|---|---|---|---|---|---|---|
| PandaGPT | 21.9 | 25.3 | 24.8 | 26.0 | 24.5 | 23.9 | 24.4 | 1.4 |
| Unified-IO 2 | 22.7 | 22.4 | 25.1 | 24.3 | 25.8 | 22.4 | 23.8 | 1.5 |
| Unified-IO 2 XL | 22.3 | 24.5 | 28.8 | 22.1 | 26.0 | 32.7 | 26.1 | 4.1 |
| Unified-IO 2 XXL | 24.3 | 27.4 | 25.3 | 29.6 | 25.2 | 34.4 | 27.7 | 3.8 |
| VideoLLaMA 2 | 31.0 | 25.0 | 27.7 | 25.9 | 25.8 | 39.8 | 29.2 | 5.6 |
| VITA | 31.1 | 25.1 | 26.1 | 24.6 | 27.6 | 41.7 | 29.4 | 6.5 |
| Baichuan Omni 1.5 | 27.0 | 25.2 | 23.9 | 26.9 | 25.0 | 52.2 | 30.0 | 10.9 |
| EchoInk-R1 | 38.2 | 26.2 | 38.6 | 31.1 | 26.9 | 61.6 | 37.1 | 13.1 |
| Qwen2.5-Omni | 26.9 | 28.7 | 36.6 | 25.6 | 25.3 | 50.8 | 32.3 | 10.0 |
| Gemini 1.5 Pro | 37.1 | 27.2 | 31.0 | 47.3 | 24.5 | 55.7 | 37.1 | 12.2 |
| Gemini 2.0 Flash | 51.8 | 30.8 | 38.6 | 48.0 | 27.4 | 72.0 | 44.8 | 16.3 |
| Gemini 2.5 Flash | 48.8 | 30.9 | 39.6 | 51.4 | 38.0 | 74.6 | 48.6 | 13.9 |
| Gemini 2.5 Pro | **76.4** | **54.4** | **57.7** | **55.4** | **50.6** | **70.6** | **60.8** | 10.3 |
| Human | 90.0 | 85.0 | 86.7 | 91.7 | 83.3 | 96.7 | 88.9 | 4.9 |

**Task 4 - Linguistic**

| Model | A↦T | A↦V | T↦A | T↦V | V↦A | V↦T | Avg. | Std. |
|---|---|---|---|---|---|---|---|---|
| PandaGPT | 28.0 | 24.3 | 20.7 | 24.7 | 24.3 | 31.3 | 25.5 | 3.6 |
| Unified-IO 2 | 32.4 | 27.5 | 27.6 | 27.9 | 25.2 | 41.7 | 30.4 | 6.0 |
| Unified-IO 2 XL | 34.4 | 31.7 | 24.5 | 28.8 | 23.6 | 41.8 | 30.8 | 6.8 |
| Unified-IO 2 XXL | 39.9 | 23.0 | 25.5 | 30.1 | 22.3 | 46.6 | 31.2 | 9.9 |
| VideoLLaMA 2 | 50.3 | 25.2 | 24.2 | 25.2 | 24.1 | 71.2 | 36.7 | 19.8 |
| VITA | 52.2 | 26.8 | 47.1 | 29.9 | 47.9 | 72.5 | 46.1 | 16.6 |
| Baichuan Omni 1.5 | 77.0 | 45.7 | 65.8 | 51.8 | 58.7 | 77.6 | 62.8 | 13.1 |
| EchoInk-R1 | 86.0 | 57.4 | 74.6 | 64.4 | 70.1 | 87.3 | 73.3 | 11.8 |
| Qwen2.5-Omni | 85.6 | 61.8 | 73.6 | 64.6 | 71.5 | 87.5 | 74.1 | 10.6 |
| Gemini 1.5 Pro | **86.2** | 52.4 | 72.3 | 68.7 | 70.7 | 85.5 | 72.6 | 12.0 |
| Gemini 2.0 Flash | 83.6 | 57.5 | 68.6 | 67.3 | 60.9 | 83.4 | 70.2 | 11.1 |
| Gemini 2.5 Flash | 84.1 | **68.3** | 70.9 | 66.8 | 64.4 | 84.4 | 73.1 | 8.9 |
| Gemini 2.5 Pro | 84.9 | 67.5 | **75.5** | **76.1** | 65.8 | **91.4** | **76.8** | 9.9 |
| Human | 89.2 | 96.7 | 97.5 | 93.3 | 91.7 | 95.0 | 93.9 | 2.8 |

**Task 5 - External Knowledge**

| Model | A↦T | A↦V | T↦A | T↦V | V↦A | V↦T | Avg. | Std. |
|---|---|---|---|---|---|---|---|---|
| PandaGPT | 22.8 | 23.9 | 22.6 | 23.6 | 23.3 | 22.6 | 23.1 | 0.5 |
| Unified-IO 2 | 28.2 | 24.2 | 25.8 | 27.1 | 25.3 | 29.9 | 26.8 | 2.1 |
| Unified-IO 2 XL | 31.9 | 27.9 | 26.2 | 28.6 | 29.5 | 32.9 | 29.5 | 2.5 |
| Unified-IO 2 XXL | 38.4 | 23.9 | 35.3 | 39.5 | 29.7 | 37.4 | 34.0 | 6.0 |
| VideoLLaMA 2 | 42.9 | 25.5 | 23.9 | 27.0 | 24.4 | 76.9 | 36.8 | 20.9 |
| VITA | 38.5 | 24.1 | 24.2 | 24.7 | 22.6 | 61.2 | 32.6 | 15.2 |
| Baichuan Omni 1.5 | 54.3 | 43.9 | 41.9 | 82.9 | 33.7 | 83.2 | 56.7 | 21.5 |
| EchoInk-R1 | 83.3 | 60.4 | 72.7 | 83.6 | 56.6 | 83.3 | 73.3 | 12.3 |
| Qwen2.5-Omni | 83.0 | 59.2 | 70.7 | 82.5 | 58.6 | 83.2 | 72.8 | 11.8 |
| Gemini 1.5 Pro | 62.3 | 52.5 | 70.2 | 88.8 | 52.3 | 90.3 | 69.4 | 17.0 |
| Gemini 2.0 Flash | 81.2 | 68.3 | 70.5 | 93.1 | 61.3 | 94.2 | 78.1 | 13.6 |
| Gemini 2.5 Flash | 82.0 | 72.2 | 81.7 | 93.9 | 74.5 | 92.7 | 82.8 | 9.0 |
| Gemini 2.5 Pro | **90.3** | **82.5** | **88.2** | **94.6** | **84.8** | **95.1** | **89.3** | 5.1 |
| Human | 96.7 | 88.3 | 98.3 | 98.3 | 85.0 | 96.7 | 93.9 | 5.8 |

# C TASK SPECIFIED MODEL PERFORMANCE

## C.1 TASK 1: PERCEPTUAL TASK

Table 6: T1 (Perception) Results

| Model | Task | Perception Task | | | | |
|---|---|---|---|---|---|---|
| Model | Task | General | General - Hard | Scene | Instruments | Instruments-multi |
| Gemini 2.5 Pro | Audio ↦ Text | 81.05 | 71.39 | 67.20 | 47.75 | 44.09 |
| | Audio ↦ Vision | 76.26 | 65.25 | 64.60 | 44.30 | 36.60 |
| | Text ↦ Audio | 79.95 | 79.22 | 75.05 | 59.05 | 49.30 |
| | Text ↦ Vision | 98.90 | 97.87 | 90.80 | 97.90 | 99.80 |
| | Vision ↦ Audio | 88.73 | 79.35 | 84.40 | 61.92 | 48.79 |
| | Vision ↦ Text | 98.37 | 97.50 | 95.00 | 97.19 | 99.80 |

**Table 6 – continued from previous page**

| Model | | Perception Task | | | | |
|---|---|---|---|---|---|---|
| Model | Task | General | General - Hard | Scene | Instruments | Instruments-multi |
| Gemini 2.5 Flash | Audio ↦ Text | 81.00 | 50.00 | 51.01 | 45.82 | 35.27 |
| | Audio ↦ Vision | 62.63 | 50.39 | 47.60 | 30.99 | 29.92 |
| | Text ↦ Audio | 79.80 | 59.13 | 57.34 | 37.90 | 32.99 |
| | Text ↦ Vision | 98.96 | 91.45 | 90.20 | 96.50 | 99.74 |
| | Vision ↦ Audio | 82.10 | 60.59 | 67.54 | 39.80 | 29.92 |
| | Vision ↦ Text | 98.39 | 96.62 | 92.60 | 89.88 | 97.27 |
| Gemini 2.0 Flash | Audio ↦ Text | 81.10 | 62.07 | 54.00 | 47.05 | 39.80 |
| | Audio ↦ Vision | 67.45 | 51.68 | 49.80 | 31.50 | 24.80 |
| | Text ↦ Audio | 79.95 | 60.64 | 53.80 | 38.80 | 37.60 |
| | Text ↦ Vision | 98.95 | 91.45 | 80.40 | 96.90 | 95.60 |
| | Vision ↦ Audio | 82.50 | 66.45 | 53.00 | 37.90 | 35.40 |
| | Vision ↦ Text | 96.95 | 90.22 | 84.80 | 96.70 | 98.40 |
| Gemini 1.5 Pro | Audio ↦ Text | 80.90 | 36.38 | 29.20 | 30.93 | 27.40 |
| | Audio ↦ Vision | 34.35 | 30.00 | 28.40 | 23.60 | 23.00 |
| | Text ↦ Audio | 80.25 | 45.88 | 41.80 | 31.10 | 26.20 |
| | Text ↦ Vision | 98.75 | 95.88 | 89.40 | 98.10 | 97.00 |
| | Vision ↦ Audio | 41.85 | 34.38 | 31.80 | 27.70 | 25.00 |
| | Vision ↦ Text | 95.10 | 94.62 | 87.80 | 98.90 | 100.00 |
| Qwen2.5 Omni | Audio ↦ Text | 80.00 | 74.50 | 79.20 | 69.37 | 61.40 |
| | Audio ↦ Vision | 71.10 | 54.30 | 59.80 | 58.30 | 51.80 |
| | Text ↦ Audio | 81.20 | 69.90 | 78.80 | 67.40 | 48.80 |
| | Text ↦ Vision | 94.90 | 87.70 | 89.60 | 90.80 | 92.80 |
| | Vision ↦ Audio | 83.90 | 68.50 | 61.20 | 68.30 | 60.60 |
| | Vision ↦ Text | 97.50 | 87.00 | 88.00 | 91.80 | 95.80 |
| EchoInk | Audio ↦ Text | 87.55 | 74.80 | 77.10 | 68.20 | 63.00 |
| | Audio ↦ Vision | 74.60 | 58.40 | 49.00 | 58.20 | 52.10 |
| | Text ↦ Audio | 84.57 | 66.40 | 79.40 | 69.14 | 46.80 |
| | Text ↦ Vision | 95.00 | 91.80 | 88.40 | 89.78 | 92.80 |
| | Vision ↦ Audio | 82.80 | 68.80 | 60.40 | 69.14 | 57.52 |
| | Vision ↦ Text | 96.00 | 95.20 | 87.80 | 92.38 | 95.79 |
| Baichuan Omni 1.5 | Audio ↦ Text | 55.85 | 44.05 | 46.40 | 36.44 | 31.00 |
| | Audio ↦ Vision | 44.45 | 37.43 | 43.20 | 29.60 | 26.60 |
| | Text ↦ Audio | 63.80 | 50.90 | 53.60 | 32.80 | 27.00 |
| | Text ↦ Vision | 97.35 | 88.10 | 81.20 | 84.50 | 88.00 |
| | Vision ↦ Audio | 68.25 | 53.75 | 58.80 | 38.40 | 32.20 |
| | Vision ↦ Text | 95.90 | 87.12 | 86.80 | 92.70 | 90.80 |
| VideoLLaMA 2 | Audio ↦ Text | 86.84 | 76.85 | 77.26 | 75.86 | 56.89 |
| | Audio ↦ Vision | 26.82 | 26.82 | 24.45 | 29.26 | 25.82 |
| | Text ↦ Audio | 30.69 | 28.21 | 28.89 | 28.45 | 25.07 |
| | Text ↦ Vision | 25.49 | 27.49 | 26.03 | 29.25 | 25.89 |
| | Vision ↦ Audio | 29.28 | 27.47 | 25.30 | 29.26 | 21.04 |
| | Vision ↦ Text | 97.05 | 91.48 | 87.45 | 89.40 | 92.23 |
| VITA | Audio ↦ Text | 43.30 | 32.99 | 39.18 | 39.18 | 30.93 |
| | Audio ↦ Vision | 22.68 | 20.62 | 28.87 | 28.87 | 25.77 |
| | Text ↦ Audio | 28.96 | 24.24 | 24.92 | 28.28 | 28.62 |
| | Text ↦ Vision | 20.62 | 25.77 | 31.96 | 21.65 | 18.56 |
| | Vision ↦ Audio | 23.57 | 29.29 | 24.92 | 25.93 | 28.28 |
| | Vision ↦ Text | 64.95 | 73.20 | 58.76 | 74.23 | 74.23 |
| Unified IO 2 | Audio ↦ Text | 49.05 | 45.26 | 32.04 | 26.46 | 24.44 |
| | Audio ↦ Vision | 27.00 | 26.84 | 30.65 | 19.40 | 22.45 |
| | Text ↦ Audio | 26.68 | 25.86 | 25.27 | 27.64 | 26.20 |
| | Text ↦ Vision | 73.28 | 56.44 | 72.67 | 27.26 | 49.89 |
| | Vision ↦ Audio | 27.89 | 24.09 | 43.22 | 24.05 | 26.26 |
| | Vision ↦ Text | 55.83 | 44.21 | 32.81 | 48.66 | 41.86 |
| Unified IO 2 XL | Audio ↦ Text | 76.64 | 71.68 | 57.82 | 33.68 | 26.87 |
| | Audio ↦ Vision | 28.04 | 25.21 | 34.89 | 22.29 | 29.22 |
| | Text ↦ Audio | 39.47 | 28.46 | 33.02 | 23.80 | 26.84 |
| | Text ↦ Vision | 81.89 | 68.28 | 69.87 | 22.09 | 53.29 |
| | Vision ↦ Audio | 26.46 | 24.43 | 35.05 | 24.80 | 27.03 |
| | Vision ↦ Text | 61.10 | 51.83 | 53.06 | 60.49 | 48.50 |
| Unified IO 2 XXL | Audio ↦ Text | 83.63 | 71.20 | 45.88 | 41.45 | 32.83 |
| | Audio ↦ Vision | 29.07 | 23.28 | 27.41 | 27.09 | 27.87 |
| | Text ↦ Audio | 59.87 | 44.10 | 35.40 | 27.68 | 28.09 |
| | Text ↦ Vision | 86.07 | 73.08 | 71.29 | 36.07 | 54.44 |
| | Vision ↦ Audio | 28.66 | 29.81 | 24.82 | 24.01 | 26.07 |
| | Vision ↦ Text | 53.46 | 48.64 | 40.49 | 61.85 | 46.68 |
| PandaGPT | Audio ↦ Text | 25.03 | 28.80 | 24.49 | 24.30 | 19.99 |
| | Audio ↦ Vision | 26.52 | 27.37 | 24.63 | 24.89 | 20.15 |

| Model | | Perception Task | | | | |
|-------|------|---------|-----------------|-------|-------------|-------------------|
| **Model** | **Task** | **General** | **General - Hard** | **Scene** | **Instruments** | **Instruments-multi** |
| PandaGPT | Text ↦ Audio | 25.40 | 29.65 | 24.25 | 24.77 | 20.08 |
| | Text ↦ Vision | 25.07 | 28.68 | 24.22 | 24.52 | 20.19 |
| | Vision ↦ Audio | 25.26 | 28.81 | 24.50 | 24.52 | 19.85 |
| | Vision ↦ Text | 25.26 | 28.70 | 24.64 | 24.90 | 20.05 |

The top of this table reads **Table 6 – continued from previous page**.

## C.2  TASK 2: SPATIAL REASONING

Table 7: T2 (Spatial) Task Results

| Model | | Spatial Task | | |
|-------|------|-------------|------------------|--------|
| **Model** | **Task** | **Arrangement** | **Moving Direction** | **Indoor** |
| Gemini 2.5 Pro | Audio ↦ Text | 28.82 | 69.39 | 24.87 |
| | Audio ↦ Vision | 24.73 | 40.65 | 33.38 |
| | Text ↦ Audio | 30.09 | 39.02 | 27.09 |
| | Text ↦ Vision | 95.70 | 58.85 | 72.73 |
| | Vision ↦ Audio | 29.01 | 38.10 | 23.86 |
| | Vision ↦ Text | 95.21 | 85.23 | 84.56 |
| Gemini 2.5 Flash | Audio ↦ Text | 27.53 | 83.53 | 25.64 |
| | Audio ↦ Vision | 26.54 | 36.03 | 31.61 |
| | Text ↦ Audio | 25.81 | 35.34 | 29.37 |
| | Text ↦ Vision | 91.40 | 66.44 | 55.71 |
| | Vision ↦ Audio | 27.44 | 26.44 | 26.12 |
| | Vision ↦ Text | 91.40 | 84.05 | 74.25 |
| Gemini 2.0 Flash | Audio ↦ Text | 28.82 | 82.71 | 24.10 |
| | Audio ↦ Vision | 26.45 | 37.58 | 35.38 |
| | Text ↦ Audio | 27.31 | 39.41 | 21.01 |
| | Text ↦ Vision | 67.53 | 66.99 | 34.62 |
| | Vision ↦ Audio | 25.81 | 45.78 | 28.86 |
| | Vision ↦ Text | 89.25 | 99.02 | 60.76 |
| Gemini 1.5 Pro | Audio ↦ Text | 29.25 | 57.37 | 24.87 |
| | Audio ↦ Vision | 27.10 | 32.87 | 33.59 |
| | Text ↦ Audio | 19.25 | 34.26 | 20.00 |
| | Text ↦ Vision | 64.30 | 50.80 | 39.23 |
| | Vision ↦ Audio | 23.66 | 21.82 | 25.57 |
| | Vision ↦ Text | 95.48 | 80.00 | 43.04 |
| Qwen2.5 Omni | Audio ↦ Text | 21.29 | 75.28 | 28.89 |
| | Audio ↦ Vision | 28.60 | 35.83 | 29.23 |
| | Text ↦ Audio | 20.22 | 31.52 | 28.35 |
| | Text ↦ Vision | 45.38 | 26.98 | 30.77 |
| | Vision ↦ Audio | 23.87 | 34.69 | 27.34 |
| | Vision ↦ Text | 80.86 | 81.63 | 41.01 |
| EchoInk | Audio ↦ Text | 27.79 | 61.62 | 34.34 |
| | Audio ↦ Vision | 24.97 | 25.59 | 30.98 |
| | Text ↦ Audio | 26.60 | 28.96 | 24.92 |
| | Text ↦ Vision | 46.80 | 25.59 | 29.63 |
| | Vision ↦ Audio | 24.88 | 31.31 | 27.95 |
| | Vision ↦ Text | 80.13 | 61.62 | 44.78 |
| Baichuan Omni 1.5 | Audio ↦ Text | 28.39 | 71.43 | 14.36 |
| | Audio ↦ Vision | 28.17 | 28.51 | 27.18 |
| | Text ↦ Audio | 22.37 | 27.21 | 25.57 |
| | Text ↦ Vision | 35.70 | 36.32 | 23.08 |
| | Vision ↦ Audio | 25.38 | 22.95 | 27.59 |
| | Vision ↦ Text | 71.40 | 82.95 | 29.37 |

**Table 7 – continued from previous page**

| Model | | Spatial Task | | |
|---|---|---|---|---|
| Model | Task | Arrangement | Moving Direction | Indoor |
| VideoLLaMA 2 | Audio ↦ Text | 31.40 | 62.44 | 37.76 |
| | Audio ↦ Vision | 27.40 | 27.75 | 28.22 |
| | Text ↦ Audio | 26.76 | 27.04 | 19.53 |
| | Text ↦ Vision | 27.36 | 27.01 | 28.25 |
| | Vision ↦ Audio | 25.63 | 29.28 | 20.77 |
| | Vision ↦ Text | 46.96 | 84.21 | 31.70 |
| VITA | Audio ↦ Text | 29.90 | 77.32 | 19.59 |
| | Audio ↦ Vision | 30.93 | 26.80 | 28.87 |
| | Text ↦ Audio | 23.23 | 25.59 | 25.00 |
| | Text ↦ Vision | 29.90 | 31.96 | 30.93 |
| | Vision ↦ Audio | 24.92 | 25.59 | 24.66 |
| | Vision ↦ Text | 57.73 | 55.67 | 43.30 |
| Unified IO 2 | Audio ↦ Text | 23.03 | 20.47 | 34.40 |
| | Audio ↦ Vision | 21.98 | 17.20 | 22.89 |
| | Text ↦ Audio | 23.50 | 20.69 | 22.87 |
| | Text ↦ Vision | 25.63 | 24.03 | 25.22 |
| | Vision ↦ Audio | 24.09 | 17.32 | 27.86 |
| | Vision ↦ Text | 28.60 | 24.10 | 21.49 |
| Unified IO 2 XL | Audio ↦ Text | 23.09 | 28.42 | 22.88 |
| | Audio ↦ Vision | 22.20 | 20.09 | 26.75 |
| | Text ↦ Audio | 24.82 | 22.92 | 29.70 |
| | Text ↦ Vision | 24.18 | 24.25 | 29.56 |
| | Vision ↦ Audio | 24.12 | 21.78 | 32.17 |
| | Vision ↦ Text | 27.41 | 24.10 | 21.93 |
| Unified IO 2 XXL | Audio ↦ Text | 22.58 | 30.07 | 36.18 |
| | Audio ↦ Vision | 24.54 | 24.02 | 22.37 |
| | Text ↦ Audio | 25.85 | 38.11 | 28.71 |
| | Text ↦ Vision | 25.39 | 30.02 | 21.10 |
| | Vision ↦ Audio | 25.45 | 28.33 | 34.77 |
| | Vision ↦ Text | 30.36 | 30.91 | 30.80 |
| PandaGPT | Audio ↦ Text | 25.42 | 25.62 | 25.44 |
| | Audio ↦ Vision | 27.22 | 25.63 | 26.91 |
| | Text ↦ Audio | 27.06 | 25.58 | 25.27 |
| | Text ↦ Vision | 27.01 | 25.95 | 28.57 |
| | Vision ↦ Audio | 27.16 | 25.53 | 24.57 |
| | Vision ↦ Text | 21.19 | 25.72 | 22.34 |

## C.3 TASK 3: TEMPORAL REASONING

Table 8: T3 (Temporal) Task Results

| Model | | Temporal Task | | |
|---|---|---|---|---|
| Model | Task | Order | Counting | Calculation |
| Gemini 2.5 Pro | Audio ↦ Text | 96.18 | 57.36 | 75.78 |
| | Audio ↦ Vision | 95.38 | 37.88 | 29.87 |
| | Text ↦ Audio | 95.39 | 50.00 | 27.60 |
| | Text ↦ Vision | 99.80 | 35.85 | 30.63 |
| | Vision ↦ Audio | 96.35 | 34.70 | 20.65 |
| | Vision ↦ Text | 99.80 | 40.58 | 71.46 |
| Gemini 2.5 Flash | Audio ↦ Text | 41.40 | 49.60 | 55.40 |
| | Audio ↦ Vision | 58.99 | 33.07 | 26.88 |
| | Text ↦ Audio | 61.00 | 29.40 | 27.00 |
| | Text ↦ Vision | 99.15 | 29.45 | 25.51 |

**Table 8 – continued from previous page**

| Model | | Temporal Task | | |
|---|---|---|---|---|
| **Model** | **Task** | **Order** | **Counting** | **Calculation** |
| | Vision $\mapsto$ Audio | 63.39 | 26.58 | 24.13 |
| | Vision $\mapsto$ Text | 99.20 | 53.37 | 71.22 |
| Gemini 2.0 Flash | Audio $\mapsto$ Text | 43.60 | 52.60 | 59.20 |
| | Audio $\mapsto$ Vision | 33.40 | 30.17 | 28.93 |
| | Text $\mapsto$ Audio | 61.40 | 28.80 | 25.60 |
| | Text $\mapsto$ Vision | 81.40 | 33.33 | 29.16 |
| | Vision $\mapsto$ Audio | 33.40 | 28.22 | 20.65 |
| | Vision $\mapsto$ Text | 99.20 | 57.87 | 58.90 |
| Gemini 1.5 Pro | Audio $\mapsto$ Text | 34.40 | 30.00 | 47.00 |
| | Audio $\mapsto$ Vision | 32.00 | 24.44 | 25.10 |
| | Text $\mapsto$ Audio | 38.60 | 30.20 | 24.20 |
| | Text $\mapsto$ Vision | 82.00 | 33.88 | 26.14 |
| | Vision $\mapsto$ Audio | 27.60 | 23.87 | 21.88 |
| | Vision $\mapsto$ Text | 98.40 | 25.26 | 43.56 |
| Qwen2.5 Omni | Audio $\mapsto$ Text | 28.20 | 25.80 | 26.60 |
| | Audio $\mapsto$ Vision | 34.80 | 22.22 | 28.96 |
| | Text $\mapsto$ Audio | 63.80 | 19.40 | 26.60 |
| | Text $\mapsto$ Vision | 24.80 | 22.90 | 28.96 |
| | Vision $\mapsto$ Audio | 26.40 | 23.57 | 25.93 |
| | Vision $\mapsto$ Text | 85.00 | 41.41 | 25.93 |
| EchoInk | Audio $\mapsto$ Text | 35.00 | 48.48 | 30.98 |
| | Audio $\mapsto$ Vision | 30.98 | 23.57 | 23.91 |
| | Text $\mapsto$ Audio | 68.69 | 21.89 | 25.25 |
| | Text $\mapsto$ Vision | 43.10 | 22.56 | 27.61 |
| | Vision $\mapsto$ Audio | 31.99 | 23.57 | 25.25 |
| | Vision $\mapsto$ Text | 93.60 | 46.80 | 44.44 |
| Baichuan Omni 1.5 | Audio $\mapsto$ Text | 23.00 | 34.40 | 23.60 |
| | Audio $\mapsto$ Vision | 23.80 | 25.74 | 25.99 |
| | Text $\mapsto$ Audio | 23.40 | 23.00 | 25.40 |
| | Text $\mapsto$ Vision | 25.80 | 26.23 | 28.77 |
| | Vision $\mapsto$ Audio | 25.20 | 28.34 | 21.47 |
| | Vision $\mapsto$ Text | 70.20 | 53.18 | 33.13 |
| VideoLLaMA 2 | Audio $\mapsto$ Text | 25.82 | 35.90 | 31.23 |
| | Audio $\mapsto$ Vision | 25.23 | 25.80 | 24.03 |
| | Text $\mapsto$ Audio | 34.29 | 22.09 | 26.70 |
| | Text $\mapsto$ Vision | 26.66 | 26.06 | 24.90 |
| | Vision $\mapsto$ Audio | 27.03 | 23.64 | 26.67 |
| | Vision $\mapsto$ Text | 50.40 | 32.44 | 36.67 |
| VITA | Audio $\mapsto$ Text | 26.26 | 38.14 | 28.87 |
| | Audio $\mapsto$ Vision | 16.49 | 31.17 | 27.52 |
| | Text $\mapsto$ Audio | 26.80 | 27.61 | 23.91 |
| | Text $\mapsto$ Vision | 22.68 | 25.62 | 25.58 |
| | Vision $\mapsto$ Audio | 28.62 | 26.71 | 27.59 |
| | Vision $\mapsto$ Text | 42.27 | 49.66 | 33.10 |
| Unified IO 2 | Audio $\mapsto$ Text | 24.28 | 18.25 | 25.44 |
| | Audio $\mapsto$ Vision | 21.50 | 22.61 | 23.03 |
| | Text $\mapsto$ Audio | 30.02 | 23.46 | 21.89 |
| | Text $\mapsto$ Vision | 25.25 | 24.85 | 22.86 |
| | Vision $\mapsto$ Audio | 25.46 | 26.29 | 25.65 |
| | Vision $\mapsto$ Text | 27.68 | 16.25 | 23.37 |
| Unified IO 2 XL | Audio $\mapsto$ Text | 24.65 | 24.63 | 17.47 |
| | Audio $\mapsto$ Vision | 26.03 | 30.21 | 17.39 |
| | Text $\mapsto$ Audio | 27.52 | 28.83 | 30.02 |
| | Text $\mapsto$ Vision | 25.09 | 19.16 | 22.17 |
| | Vision $\mapsto$ Audio | 22.64 | 24.92 | 30.57 |

**Table 8 – continued from previous page**

| Model | | Temporal Task | | |
|---|---|---|---|---|
| **Model** | **Task** | **Order** | **Counting** | **Calculation** |
| | Vision ↦ Text | 37.30 | 36.42 | 24.44 |
| Unified IO 2 XXL | Audio ↦ Text | 24.41 | 26.81 | 21.62 |
| | Audio ↦ Vision | 25.26 | 29.68 | 27.17 |
| | Text ↦ Audio | 28.83 | 22.43 | 24.61 |
| | Text ↦ Vision | 23.70 | 37.78 | 27.37 |
| | Vision ↦ Audio | 23.63 | 24.69 | 27.28 |
| | Vision ↦ Text | 41.69 | 38.50 | 22.95 |
| Panda | Audio ↦ Text | 25.85 | 16.77 | 23.17 |
| | Audio ↦ Vision | 26.06 | 22.60 | 27.31 |
| | Text ↦ Audio | 25.72 | 22.81 | 25.80 |
| | Text ↦ Vision | 26.31 | 22.77 | 29.02 |
| | Vision ↦ Audio | 26.10 | 22.77 | 24.59 |
| | Vision ↦ Text | 25.51 | 22.94 | 23.37 |

## C.4 TASK 4: LINGUISTIC TASK

Table 9: T4 Linguistic Task Results

| Model | | Linguistic Task | | |
|---|---|---|---|---|
| **Model** | **Task** | **Recognition** | **Translation** | **Emotion** |
| Gemini 2.5 Pro | Audio ↦ Text | 97.16 | 96.58 | 60.86 |
| | Audio ↦ Vision | 91.65 | 67.95 | 42.75 |
| | Text ↦ Audio | 80.35 | 81.62 | 64.51 |
| | Text ↦ Vision | 93.58 | 67.38 | 67.31 |
| | Vision ↦ Audio | 80.81 | 73.22 | 43.43 |
| | Vision ↦ Text | 99.54 | 100.00 | 74.54 |
| Gemini 2.5 Flash | Audio ↦ Text | 94.05 | 97.44 | 60.86 |
| | Audio ↦ Vision | 68.01 | 93.30 | 43.67 |
| | Text ↦ Audio | 76.92 | 81.34 | 54.43 |
| | Text ↦ Vision | 72.88 | 67.24 | 60.14 |
| | Vision ↦ Audio | 74.95 | 72.93 | 45.22 |
| | Vision ↦ Text | 99.40 | 96.72 | 57.14 |
| Gemini 2.0 Flash | Audio ↦ Text | 92.86 | 97.29 | 60.57 |
| | Audio ↦ Vision | 68.30 | 67.66 | 36.43 |
| | Text ↦ Audio | 69.79 | 81.20 | 54.86 |
| | Text ↦ Vision | 73.92 | 67.66 | 60.43 |
| | Vision ↦ Audio | 66.52 | 73.08 | 43.00 |
| | Vision ↦ Text | 96.43 | 97.15 | 56.71 |
| Gemini 1.5 Pro | Audio ↦ Text | 94.94 | 97.15 | 60.43 |
| | Audio ↦ Vision | 73.96 | 46.72 | 36.57 |
| | Text ↦ Audio | 83.33 | 80.91 | 52.57 |
| | Text ↦ Vision | 76.93 | 66.81 | 62.43 |
| | Vision ↦ Audio | 80.80 | 92.02 | 39.20 |
| | Vision ↦ Text | 96.73 | 96.44 | 63.29 |
| Qwen2.5 Omni | Audio ↦ Text | 94.64 | 96.72 | 65.29 |
| | Audio ↦ Vision | 62.95 | 73.36 | 48.94 |
| | Text ↦ Audio | 81.25 | 86.75 | 52.71 |
| | Text ↦ Vision | 65.03 | 69.09 | 59.79 |
| | Vision ↦ Audio | 82.44 | 88.60 | 43.57 |
| | Vision ↦ Text | 97.17 | 97.72 | 67.57 |
| EchoInk | Audio ↦ Text | 92.93 | 95.96 | 69.02 |
| | Audio ↦ Vision | 64.98 | 71.38 | 35.69 |
| | Text ↦ Audio | 80.47 | 81.48 | 61.95 |

**Table 9 – continued from previous page**

| Model | | Linguistic Task | | |
|---|---|---|---|---|
| **Model** | **Task** | **Recognition** | **Translation** | **Emotion** |
| | Text $\mapsto$ Vision | 68.35 | 67.68 | 57.24 |
| | Vision $\mapsto$ Audio | 81.48 | 85.86 | 43.10 |
| | Vision $\mapsto$ Text | 96.63 | 97.31 | 68.01 |
| Baichuan Omni 1.5 | Audio $\mapsto$ Text | 87.05 | 96.01 | 48.00 |
| | Audio $\mapsto$ Vision | 55.36 | 56.55 | 25.25 |
| | Text $\mapsto$ Audio | 64.29 | 84.94 | 48.29 |
| | Text $\mapsto$ Vision | 55.95 | 52.56 | 46.99 |
| | Vision $\mapsto$ Audio | 65.03 | 84.06 | 27.14 |
| | Vision $\mapsto$ Text | 92.56 | 96.72 | 43.43 |
| VideoLLaMA 2 | Audio $\mapsto$ Text | 69.04 | 67.40 | 14.48 |
| | Audio $\mapsto$ Vision | 24.82 | 26.00 | 24.68 |
| | Text $\mapsto$ Audio | 22.82 | 22.02 | 27.68 |
| | Text $\mapsto$ Vision | 25.03 | 25.80 | 24.65 |
| | Vision $\mapsto$ Audio | 24.07 | 23.25 | 25.01 |
| | Vision $\mapsto$ Text | 83.86 | 86.80 | 43.00 |
| VITA | Audio $\mapsto$ Text | 39.18 | 73.20 | 44.33 |
| | Audio $\mapsto$ Vision | 24.74 | 24.74 | 30.93 |
| | Text $\mapsto$ Audio | 39.73 | 55.56 | 46.13 |
| | Text $\mapsto$ Vision | 30.93 | 25.77 | 32.99 |
| | Vision $\mapsto$ Audio | 53.87 | 61.95 | 27.95 |
| | Vision $\mapsto$ Text | 86.60 | 88.66 | 42.27 |
| Unified IO 2 | Audio $\mapsto$ Text | 62.01 | 14.06 | 21.05 |
| | Audio $\mapsto$ Vision | 35.66 | 20.90 | 25.83 |
| | Text $\mapsto$ Audio | 26.60 | 26.36 | 29.85 |
| | Text $\mapsto$ Vision | 25.89 | 26.00 | 31.82 |
| | Vision $\mapsto$ Audio | 24.24 | 25.14 | 26.27 |
| | Vision $\mapsto$ Text | 66.06 | 18.90 | 40.01 |
| Unified IO 2 XL | Audio $\mapsto$ Text | 69.63 | 17.26 | 16.29 |
| | Audio $\mapsto$ Vision | 45.46 | 26.28 | 23.28 |
| | Text $\mapsto$ Audio | 27.82 | 23.75 | 21.90 |
| | Text $\mapsto$ Vision | 30.65 | 25.26 | 30.47 |
| | Vision $\mapsto$ Audio | 25.07 | 23.70 | 21.88 |
| | Vision $\mapsto$ Text | 75.27 | 23.23 | 27.02 |
| Unified IO 2 XXL | Audio $\mapsto$ Text | 72.67 | 17.63 | 29.46 |
| | Audio $\mapsto$ Vision | 18.23 | 27.43 | 23.24 |
| | Text $\mapsto$ Audio | 23.04 | 25.97 | 27.47 |
| | Text $\mapsto$ Vision | 31.09 | 27.84 | 31.42 |
| | Vision $\mapsto$ Audio | 19.43 | 23.31 | 24.06 |
| | Vision $\mapsto$ Text | 78.04 | 26.88 | 34.88 |
| PandaGPT | Audio $\mapsto$ Text | 27.12 | 28.83 | 28.03 |
| | Audio $\mapsto$ Vision | 22.38 | 22.23 | 28.23 |
| | Text $\mapsto$ Audio | 22.06 | 18.88 | 21.20 |
| | Text $\mapsto$ Vision | 22.10 | 24.96 | 27.04 |
| | Vision $\mapsto$ Audio | 22.55 | 22.40 | 27.99 |
| | Vision $\mapsto$ Text | 33.96 | 32.67 | 27.20 |

## C.5 TASK 5: EXTERNAL KNOWLEDGE

Table 10: T5 (External) Task Results

| Model | | External Task | | |
|---|---|---|---|---|
| **Model** | **Task** | **Genre** | **Movie** | **Singer** |
| Gemini 2.5 Pro | Audio ↦ Text | 83.28 | 93.00 | 94.67 |
| | Audio ↦ Vision | 74.80 | 89.90 | 82.67 |
| | Text ↦ Audio | 78.16 | 94.50 | 91.95 |
| | Text ↦ Vision | 85.76 | 97.99 | 100.00 |
| | Vision ↦ Audio | 72.42 | 92.00 | 90.00 |
| | Vision ↦ Text | 88.95 | 96.45 | 100.00 |
| Gemini 2.5 Flash | Audio ↦ Text | 83.78 | 93.00 | 69.13 |
| | Audio ↦ Vision | 63.36 | 82.41 | 70.92 |
| | Text ↦ Audio | 78.56 | 90.45 | 76.00 |
| | Text ↦ Vision | 85.00 | 97.99 | 98.67 |
| | Vision ↦ Audio | 63.96 | 88.32 | 71.33 |
| | Vision ↦ Text | 86.34 | 98.00 | 93.71 |
| Gemini 2.0 Flash | Audio ↦ Text | 83.50 | 88.00 | 72.00 |
| | Audio ↦ Vision | 62.40 | 86.50 | 56.00 |
| | Text ↦ Audio | 78.46 | 82.50 | 50.67 |
| | Text ↦ Vision | 84.50 | 98.00 | 96.67 |
| | Vision ↦ Audio | 66.43 | 79.50 | 38.00 |
| | Vision ↦ Text | 87.50 | 95.00 | 100.00 |
| Gemini 1.5 Pro | Audio ↦ Text | 61.70 | 78.00 | 47.33 |
| | Audio ↦ Vision | 42.90 | 74.50 | 40.00 |
| | Text ↦ Audio | 63.53 | 84.50 | 62.67 |
| | Text ↦ Vision | 82.10 | 95.50 | 88.67 |
| | Vision ↦ Audio | 45.59 | 74.00 | 37.33 |
| | Vision ↦ Text | 87.10 | 95.00 | 88.67 |
| Qwen2.5 Omni | Audio ↦ Text | 89.50 | 79.50 | 80.00 |
| | Audio ↦ Vision | 61.40 | 67.50 | 48.67 |
| | Text ↦ Audio | 85.65 | 70.50 | 56.00 |
| | Text ↦ Vision | 74.20 | 94.50 | 78.67 |
| | Vision ↦ Audio | 81.82 | 60.50 | 33.33 |
| | Vision ↦ Text | 79.00 | 92.50 | 78.00 |
| EchoInk | Audio ↦ Text | 87.54 | 82.50 | 80.00 |
| | Audio ↦ Vision | 61.95 | 68.00 | 51.33 |
| | Text ↦ Audio | 84.51 | 73.00 | 60.67 |
| | Text ↦ Vision | 77.78 | 93.00 | 80.00 |
| | Vision ↦ Audio | 62.63 | 64.50 | 42.67 |
| | Vision ↦ Text | 79.12 | 93.50 | 77.33 |
| Baichuan Omni 1.5 | Audio ↦ Text | 65.60 | 56.00 | 41.33 |
| | Audio ↦ Vision | 45.30 | 54.50 | 32.00 |
| | Text ↦ Audio | 25.75 | 60.00 | 40.00 |
| | Text ↦ Vision | 77.00 | 94.50 | 77.33 |
| | Vision ↦ Audio | 27.15 | 46.50 | 27.33 |
| | Vision ↦ Text | 81.20 | 94.50 | 74.00 |
| VideoLLaMA 2 | Audio ↦ Text | 62.60 | 26.59 | 39.38 |
| | Audio ↦ Vision | 26.23 | 23.56 | 26.72 |
| | Text ↦ Audio | 24.85 | 21.59 | 25.34 |
| | Text ↦ Vision | 26.40 | 28.55 | 26.09 |
| | Vision ↦ Audio | 25.67 | 23.56 | 24.10 |
| | Vision ↦ Text | 68.27 | 80.55 | 82.02 |
| VITA | Audio ↦ Text | 46.39 | 40.21 | 28.87 |
| | Audio ↦ Vision | 20.62 | 26.80 | 24.74 |
| | Text ↦ Audio | 21.89 | 25.50 | 25.33 |
| | Text ↦ Vision | 20.62 | 31.96 | 21.65 |
| | Vision ↦ Audio | 23.23 | 22.00 | 22.67 |
| | Vision ↦ Text | 47.42 | 81.44 | 54.64 |

**Table 10 – continued from previous page**

| Model | | External Task | | |
| Model | Task | Genre | Movie | Singer |
|---|---|---|---|---|
| Unified IO 2 | Audio ↦ Text | 31.83 | 22.53 | 30.09 |
| | Audio ↦ Vision | 22.30 | 29.03 | 21.40 |
| | Text ↦ Audio | 26.25 | 24.51 | 26.71 |
| | Text ↦ Vision | 34.46 | 26.03 | 20.71 |
| | Vision ↦ Audio | 25.45 | 20.57 | 30.01 |
| | Vision ↦ Text | 27.90 | 34.59 | 27.33 |
| Unified IO 2 XL | Audio ↦ Text | 36.80 | 27.52 | 31.40 |
| | Audio ↦ Vision | 29.23 | 29.09 | 25.40 |
| | Text ↦ Audio | 24.12 | 25.09 | 29.34 |
| | Text ↦ Vision | 34.41 | 24.57 | 26.68 |
| | Vision ↦ Audio | 26.51 | 32.55 | 29.41 |
| | Vision ↦ Text | 24.86 | 35.76 | 38.05 |
| Unified IO 2 XXL | Audio ↦ Text | 57.68 | 22.71 | 34.70 |
| | Audio ↦ Vision | 26.83 | 20.56 | 24.42 |
| | Text ↦ Audio | 47.92 | 26.52 | 31.43 |
| | Text ↦ Vision | 51.85 | 24.01 | 42.75 |
| | Vision ↦ Audio | 25.06 | 30.57 | 33.40 |
| | Vision ↦ Text | 28.20 | 36.55 | 47.36 |
| Panda | Audio ↦ Text | 25.77 | 21.32 | 21.24 |
| | Audio ↦ Vision | 25.74 | 24.49 | 21.42 |
| | Text ↦ Audio | 22.11 | 25.18 | 20.37 |
| | Text ↦ Vision | 24.63 | 24.64 | 21.40 |
| | Vision ↦ Audio | 23.93 | 24.58 | 21.29 |
| | Vision ↦ Text | 26.32 | 21.07 | 20.39 |

## D  EVALUATION COST

We provide a detailed evaluation cost section as a reference of usage. We evaluate on the full version (60k sample) of XModBench, API-based models we test **Gemini 2.5 Pro**, we report the *token usage* for evaluating the overall benchmark and each task family . For open-source models we report **Qwen2.5-Omni**, we report the *evaluation runing time*, using with eight `A6000` GPUs and each GPU run one process.

Table 11: Evaluation cost estimation for models across the five task families and the full benchmark.

| Model | Perc. | Spat. | Temp. | Ling. | Knwl. | Total |
|---|---|---|---|---|---|---|
| Gemini 2.5 Pro (*Token usage*) | 26.0M | 13.5M | 25.1M | 4.3M | 14.0M | 82.9M |
| Qwen2.5-Omni (*Hours*) | 6.3 | 1.4 | 1.4 | 1.4 | 2.1 | 12.7 |

## E  INTERLEAVING VISUAL AUDIO INPUT

In the preceding experiments, we showed that omni-language models exhibit varying performance in pairwise cross-modal reasoning, particularly between vision–text and audio–text tasks. Yet, real-world multimodal scenarios are more complex: information from multiple modalities often arrives simultaneously and must be processed in an integrated manner. To address this challenge, we extend all tasks in XModBench to an audio–visual context configuration, where the question stem provides both audio and visual cues, while the candidate space remains identical to the original text-based setting.

We evaluate this dual-context setup using the Gemini series of models, which represent some of the most advanced omni-language systems available. The results, presented in Tab. 12, enable a direct comparison with the pairwise baseline and reveal how models leverage—or fail to leverage—simultaneous multimodal evidence.

Table 12: Overall performance of Gemini models under the dual-context setting (audio+visual context ↦ text). We compare with pairwise baselines (A ↦ T and V ↦ T), and report the stronger unimodal baseline $\max(A \mapsto T, V \mapsto T)$.

| Setting | Gemini 1.5 Pro | Gemini 2.0 Flash | Gemini 2.5 Pro |
|---|---|---|---|
| A ↦ T | 52.76 | 63.71 | 70.99 |
| V ↦ T | 79.92 | 85.20 | 88.60 |
| A+V ↦ T | 82.53 (**+2.61**) | 79.84 | 89.76 |

## F   HUMAN SURVEY

To evaluate human performance and establish reference baselines, we conducted a user study on a subset of **XModBench**. Participants answered multiple-choice questions under different modality configurations, with Figure 7 showing a screenshot of the interface and example questions. For each subtask, we collected responses from 10 valid participants per modality configuration.

## G   TECHINIQAL DETAILS IN TRIPLET DATA COLLECTION AND PROCESSING DATA FOR EACH SUBTASK

In this section, we provide detailed descriptions of the data sources are collected, and how each data in each modality are processed for each subtask in XModBench.

### G.1   PERCEPTUAL RECOGNITION

**General Categories.**   We utilize the VGGSound Source (VGG-SS) dataset(Chen et al., 2021; Kim et al., 2024), a large-scale video benchmark designed for sound source localization, which provides video-level annotations across diverse sound activities. The dataset covers 200 categories with approximately 5,000 video clips, where sound sources are annotated with bounding boxes to ensure clear visibility in each clip. For our benchmark, we extract a 2-second segment corresponding to the loudest audio channel as the audio input, and randomly sample a single frame from the same clip as the visual input. The activity class name serves as the textual description. To construct multiple-choice questions, four additional activity labels are randomly sampled as distractors, resulting in four candidate answers per instance. We then use Gemini 2.5-flash lite to(Comanici et al., 2025) filter if each instance if the audio and video frame is clear to be hear and the image frame and audio are all match the category name.

**Fine-grained Categories.**   This subtask uses the same pool of video clips as the General Categories setting. The difference lies in reorganizing the activity classes into eight fine-grained clusters: *Animal sounds*, *Musical instruments*, *Human activities*, *Transportation*, *Tools and utilities*, *Urban sounds*, *Human speech*, and *Natural sounds*. For each instance, we select the target activity along with four distractor activities sampled from the same fine-grained cluster. This ensures that all answer choices belong to the same semantic domain, making the recognition task more challenging and diagnostic within a coherent category group.

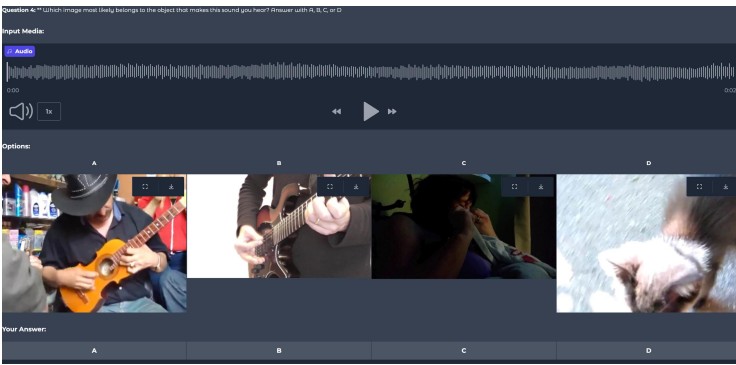

Figure 7: Sample question of human survey

**Natural Environment.** We draw data from the Landscapes dataset(Lee et al., 2022), which consists of ambient audio–video clips capturing natural outdoor scenes. Following the same selection protocol as in the General Categories task, we extract a 2-second segment from the dominant audio channel as the audio input, and randomly sample one frame from the corresponding video as the visual input. The dataset's categorical labels are used as the textual descriptions.

**Instruments.** Instrument data is collected from the Solos dataset(Montesinos et al., 2020), which contains recordings of 13 distinct instruments: violin, viola, cello, double bass, flute, oboe, clarinet, bassoon, saxophone, trumpet, horn, trombone, and tuba. We use the video frames as the visual modality, the isolated performance recordings as the audio modality, and the instrument names as textual labels.

**Instrument Composition.** We employ the URMP dataset(Li et al., 2018), a multimodal corpus designed for music performance analysis, which provides video and audio recordings of ensemble performances. For this subtask, we leverage clips containing multiple instruments playing together, using the mixture audio as input, sampled video frames as the visual modality, and instrument combination labels as text.

### G.2 SPATIAL REASONING

**2D Horizontal Arrangement.** This subtask is derived from the URMP dataset(Li et al., 2018), which contains multi-instrument ensemble recordings with annotated left-to-right spatial positions of each performer and independent audio channels per instrument. We construct multiple-choice questions by generating three distractor options through random shuffling of instrument order along the horizontal axis. For the visual modality, cropped player images are concatenated into a composite frame that preserves their spatial arrangement. For the audio modality, stereo spatialization is synthesized by assigning distinct azimuth values to each shuffled configuration and adjusting the relative channel balance using a panning algorithm (e.g., vector-base amplitude panning(Pulkki, 1997)). This design ensures that listeners can clearly perceive the relative horizontal positions of the instruments.

**3D Localization.** This subtask builds on the STARSS23 dataset(Shimada et al., 2023), which provides panoramic video with time-stamped annotations of sound source depth, azimuth, and activity. For the visual modality, we annotate sound sources with bounding boxes and generate alternative views by rotating the camera perspective to $+90°$, $180°$, and $-90°$ (positive defined as left). The corresponding videos are created through spatial cropping of frames. For the audio modality, we utilize the four-channel microphone array (MIC) recordings and simulate azimuthal rotation by first encoding the array signals into first-order Ambisonics (FOA), applying a 2D rotation matrix to the X–Y components, and decoding back into microphone signals with loudness normalization. To further enhance perceptual realism, each spatial microphone signal is additionally processed with head-related transfer functions (HRTFs) in the SOFA format(Majdak et al., 2013; Algazi et al., 2001).

**3D Movements.** This subtask is based on the Urbansas dataset(Fuentes et al., 2022), which provides street-view traffic videos with detailed audio annotations indicating vehicle types and the presence of off-screen sounds. Each clip includes labels specifying the vehicle category, whether the sound source is visible in the video, and its temporal activity. We curate video segments from this dataset and highlight the target vehicle by overlaying a red bounding box to establish clear audio–visual correspondence.

### G.3 TEMPORAL REASONING

**Event Order.** This subtask is derived from 2-second video clips in the VGGSound Source (VGG-SS) dataset(Chen et al., 2021), originally used in the Perceptual Recognition task where each clip is annotated with an activity class label. For temporal ordering, we randomly sample 3–5 clips from different classes and generate four candidate event sequences by shuffling their order. Each sequence is represented across three modalities: (i) a text description (e.g., "Event A → Event B → Event C"), (ii) a concatenated video sequence, and (iii) a concatenated audio sequence. Multiple-choice questions are formed by selecting one sequence as the correct answer and presenting the stem in one modality, while the four candidate sequences are given in another modality.

**Repetition Count.** Following the setup in(Zhang et al., 2021), this subtask focuses on counting repeated events. Visual data is generated from synthetic renderings of repeated object actions, while audio data consists of temporal patterns with clear repetitions (e.g., sequences of knocks or claps). Text prompts explicitly query the number of repetitions in either modality.

**Repetition Calculation.** Also inspired by(Zhang et al., 2021), this subtask extends beyond direct counting by requiring simple arithmetic over observed repetitions. Both audio and video are rendered with variable fre-

quencies of repeated events, while the text prompts encode arithmetic formulations that ask models to compute totals (e.g., "three knocks plus two knocks").

## G.4    Linguistic Understanding

**Linguistic Recognition.**    This subtask targets recognition of textual content across modalities. Images are collected from OCR-rendered text data(Wendler, 2024), each paired with its ground-truth transcript. Audio is generated from these transcripts using a TTS system(Guo et al., 2025), allowing for cross-modal recognition between text, vision, and speech.

**Translation.**    This subtask examines cross-lingual translation. Input sequences consist of English text with multiple-choice options in Chinese. Text data is derived from OCR-rendered images(Wendler, 2024), while translations are generated using Gemini(Team et al., 2024). Visual inputs are rendered using the OCR dataset rendering toolkit(GbotHQ, 2024), and audio is synthesized from both languages with a TTS system(Guo et al., 2025).

**Dialogue Emotion.**    This subtask focuses on multimodal emotion recognition in conversational settings. Visual data consists of face videos displaying emotional expressions extracted from multi-party dialogue clips(Chen et al., 2018; Poria et al., 2019). Each dialogue is paired with transcripts and annotated with categorical emotions (anger, disgust, fear, sadness, surprise, and joy). We filter clips to lengths between 5–30 seconds. The video data is stripped of original audio but accompanied by transcripts to enable inference of emotion from dialogue and facial expression. Audio inputs consist of the original speech tracks, and text inputs are provided as the emotion category names.

## G.5    External Knowledge

**Music Genre Classification.**    This subtask evaluates music genre recognition. We collect audio samples from the GTZAN dataset(Olteanu, 2024), covering multiple musical styles. To complement the audio, we also collect representative album cover images for each genre category.

**Movie Matching.**    This subtask requires linking multimodal cues to movie identities. We collect a set of recent films from IMDb. For the visual modality, we use official posters. To prevent trivial text matching between posters and movie titles, we use written plot summaries from IMDb as the text modality. Audio is sampled as 30-second clips from publicly available trailers on YouTube.

**Singer Identification.**    This subtask targets cross-modal recognition of popular singers. Images of singers are collected from the web, while audio consists of short clips (3–5 songs each) sampled from their publicly available music videos on YouTube. Text inputs include singer names and associated biographical metadata. We select a diverse set of internationally recognized artists, including American singers Ariana Grande, Bad Bunny, Billie Eilish, Bruno Mars, Chappell Roan, Harry Styles, and Chinese singers David Tao, Eason Chan, Faye Wong, G.E.M., and Jay Chou.

## H    LLM Usage

We used large language models (LLMs) to assist in the preparation of this paper. Their role was limited to language editing such as proofreading and rephrasing. All ideas, experiments, and analyses were conceived and conducted by the authors.