# OpenReview forum: "XModBench: Benchmarking Cross-Modal Capabilities and Consistency in Omni-Language Models"
_ICLR.cc/2026/Conference — ICLR 2026 Poster_

### Official Review · Reviewer_DEkV · 2025-10-27

**Soundness:** 3
**Presentation:** 3
**Contribution:** 3
**Rating:** 4
**Confidence:** 3

**Summary:**

This paper presents XModBench, a large-scale tri-modal benchmark for evaluating cross-modal consistency in omni-language models (OLLMs) that handle text, vision, and audio.
Unlike previous multimodal benchmarks, XModBench focuses on whether models maintain consistent reasoning across modalities.
It includes 60K multiple-choice questions spanning five task families and six modality directions, and introduces diagnostic metrics for task competence, modality disparity, and directional imbalance.
Experiments on leading OLLMs (e.g., Gemini 2.5 Pro, Qwen2.5-Omni) show that current models still lack modality-invariant reasoning, especially in spatial, temporal, and audio-related tasks.

**Strengths:**

Strengths

Novel Focus: Targets an important but underexplored problem — evaluating cross-modal consistency rather than just multimodal performance.

Comprehensive Benchmark Design: Covers six modality directions and five task families, providing a balanced and systematic tri-modal evaluation.

Insightful Diagnostics: Introduces clear metrics (modality disparity and directional imbalance) that reveal hidden biases and asymmetries in current OLLMs.

**Weaknesses:**

This paper compares model performance differences across modalities for the same question. However, it does not discuss whether such differences are caused by information loss during modality conversion — for example, when a video or audio question is converted into text, the textual description cannot fully capture the visual or auditory content.

The paper does not explore whether specific training strategies or data construction methods could help mitigate these shortcomings in cross-modal consistency.

**Questions:**

None

---

> ### Author Response · Authors · 2025-11-26
> **1. Information Loss During Modality Conversion**
>
> **Your Concern:**
>
> > (Weakness 1) The paper compares model performance across modalities for the same question, but does not analyze whether differences might be caused by information loss when converting audio/video questions into text descriptions. Text may not fully capture visual or auditory content.
>
> **Response:**
>
> We appreciate the reviewer's perspective and clarify that our design intentionally avoids information loss during modality conversion:
>
> 1. **Text queries contain all task-relevant information.** During our template-based construction, we have ensures that textual / visual / audio format can explicitly encode all necessary details — object label, spatial relations, temporal events, and Singer identities. We have removed the annotations mistakes or low quality data by human filtering (Sec. 3). And avoid similar distractor, for example audio similar cases (e.g., "woman speaking", "women singing"). Therefore, the textual version is semantically complete for answering.
>
> 2. **Human evaluation confirms zero-loss equivalence.** We also conduct the human and the result reported in Table 2 to ensure such information loss does not exist. As annotators achieve consistent performance across modalities (above 88%), indicating that the all representations preserve the essential semantics of the task specific context. Since humans exhibit near-perfect consistency across modalities, but models do not, the observed discrepancies reflect *model limitations* in cross-modal grounding rather than information loss in the benchmark design.

---

> ### Author Response · Authors · 2025-11-26
> **2.  Training Strategies or Data Construction to Improve Cross-Modal Consistency**
>
> **Your Concern:**
>
> > (Weakness 2) The paper does not discuss whether training strategies or data construction methods could mitigate cross-modal consistency shortcomings.
>
> **Response:**
>
> We agree that understanding how training strategies and data construction choices influence cross-modal consistency is an important question. In the revised manuscript, we add a dedicated discussion section (Section 5) that synthesizes our empirical findings and analyzes three factors that directly relate to the reviewer's concern: data interleaving, domain coverage, and post-training dynamics.
>
> Our benchmark suggests several directions that may help mitigate cross-modal inconsistency:
>
> - **Interleaved multimodal training.** Models trained on large-scale interleaved audio–vision–text corpora (such as Gemini and Qwen-Omni) show much stronger symmetry in modality-swap tasks. This indicates that exposing modalities in shared contexts is an effective way to stabilize cross-modal alignment.
>
> - **Broader and more balanced modality-domain coverage.** Performance gaps on non-spoken audio, music, and spatial tasks reveal that uneven domain coverage can directly harm multimodal consistency. Including a wider range of natural audio and spatial-vision data appears essential for improving robustness across task families.
>
> - **Post-training strategies that preserve cross-modal grounding.** Our comparison between EchoInk-R1 and the Qwen 2.5 omni shows that aggressive instruction tuning on unimodal or speech-focused datasets can weaken previously learned cross-modal grounding.
>
> These insights, derived from XModBench's diagnostic design, highlight not only the current limitations of multimodal models but also practical strategies for improving cross-modal consistency in future systems. We believe that such observations are largely absent from existing OLLM benchmarks, which tend to focus primarily on absolute accuracy rather than revealing the underlying factors that to perform better multimodal alignment.

---

### Official Review · Reviewer_a37s · 2025-10-29

**Soundness:** 4
**Presentation:** 3
**Contribution:** 3
**Rating:** 6
**Confidence:** 4

**Summary:**

The paper introduces XModBench, a large-scale, tri-modal (audio, vision, text) benchmark designed to evaluate omni-modal large language models (OLLMs). The core objective is to move beyond task-specific accuracy and measure cross-modal consistency—the ability of a model to produce consistent answers when the same semantic content is presented in different modalities. XModBench comprises 60,828 multiple-choice questions systematically covering five task families (perception, spatial reasoning, temporal reasoning, linguistic understanding, and external knowledge) and all six possible cross-modal directions between the three modalities. The authors use this benchmark to evaluate a range of OLLMs, including the Gemini series and several open-source models. The key findings demonstrate that even state-of-the-art models like Gemini 2.5 Pro lack true modality-invariant reasoning, exhibiting significant performance drops on spatial/temporal tasks, major disparities when inputs are switched (e.g., text vs. audio), and directional imbalances (e.g., V->T vs. T->V).

**Strengths:**

1. The paper tackles a crucial question: are OLLMs truly modality-invariant? It moves evaluation beyond simple accuracy on multimodal tasks and proposes a novel, principled method for measuring cross-modal consistency. The design, which permutes modalities for the same semantic question, is the paper's core strength.



2. The benchmark is comprehensive. It contains over 60K questions , covers 5 diverse task families (from perception to external knowledge) , and 17 subtasks. This breadth ensures that the findings are not artifacts of a single domain.



3. High-Quality Curation. The authors detail a rigorous data curation and verification process. The explicit use of "human in-the-loop verification" and multiple rounds of testing by annotators addresses major concerns about the quality and ambiguity of web-sourced or generated data .

4. Actionable Diagnostics. The paper doesn't just rank models. It provides specific, interpretable diagnostic metrics—modality disparity and directional imbalance —that allow researchers to pinpoint where and how their models are failing. The failure case analysis in Section 4.5 and Figure 6 reinforces this with qualitative examples

**Weaknesses:**

1. Accessibility and Cost: The benchmark's primary strength—its scale—is also a potential weakness for adoption. Evaluating a model on 60,828 question-answer pairs, many of which involve multiple modalities, appears to be a computationally expensive process. The paper does not mention the availability of a smaller, standardized "lite" subset for researchers with limited compute. Furthermore, no information is provided on the practical costs of evaluation, such as total token usage for API-based models (like the Gemini series) or GPU hours for open-source models. This omission could be a significant barrier to widespread adoption and reproducibility.

2. Limited Analysis of SOTA Performance: The paper's results clearly establish Gemini 2.5 Pro as the top-performing model, yet one that still has significant flaws. However, the analysis is largely limited to reporting these scores and failures. The paper would be strengthened by a deeper discussion hypothesizing why this model performs so much better on average than its open-source counterparts. Is it its training data, a specific architectural choice, or better-aligned encoders? A more in-depth analysis of the causes of SOTA performance (and its limitations) would be more impactful than just documenting the performance itself. Could be helpful if we could analyze how it was trained even with a guess.

**Questions:**

Given the impressive scale of the benchmark, have the authors considered releasing a standardized "lite" subset? A smaller, balanced subset would significantly lower the barrier to entry, allowing for more rapid experimentation and broader adoption by the research community.

Could the authors provide an estimation of the computational cost to run the full XModBench evaluation? Specifically, what is the approximate token usage (input and output) for evaluating an API-based model like Gemini, and what are the estimated GPU-hours for an open-source model?

The performance of Gemini 2.5 Pro is a key data point. While its failures on spatial/temporal tasks are clear , its overall superiority is also evident. Do the authors have any insights or hypotheses as to why this model demonstrates relatively better cross-modal consistency and overall competence compared to the other models tested?

**Details Of Ethics Concerns:**

The authors use web-sourced data without providing ethic statements.

---

> ### Author Response · Authors · 2025-11-26
> **Accessibility, Evaluation Cost, and a "Lite" Subset**
>
> **Your Concern:**
>
> > (Weakness 1) The benchmark's scale (60,828 QA pairs) may limit adoption. The paper does not mention a smaller standardized subset, nor does it provide information on computational or monetary cost (token usage for API models or GPU-hours for open-source models).
> >
> > (Question 1) Have the authors considered releasing a balanced, smaller subset for rapid experimentation?
> >
> > (Question 2) Could the authors provide approximate token usage for API-based models and GPU-hours for open-source models?
>
> **Response:**
>
> We appreciate the reviewer's emphasis on accessibility and reproducibility. In the revision, we will clarify the following points:
>
> 1. **Lite Subset.** We will release a standardized **6k-sample "XModBench-Lite"** consisting of **5 tasks** × **6 modality** configuration categories with **200** examples each setting, balanced across task families and modalities. This subset supports rapid prototyping on limited compute while retaining the benchmark's diagnostic value. The updated results for this subset will be included in Appendix A. Both the full and lite versions will be open-sourced to facilitate broader adoption.
>
> 2. **Evaluation Cost.** We will provide a detailed evaluation-cost table in the revision. For API-based models such as Gemini 2.5 Pro, we report the token usage for evaluating the full benchmark and different tasks. For open-source models such as Qwen-2.5-Omni we report the time required to complete the evaluation on one server with 8 `A6000` GPUs,
>
> **Table:** Evaluation cost estimation for models across the five task families and the full benchmark.
>
> | **Model** | **Perc.** | **Spat.** | **Temp.** | **Ling.** | **Knwl.** | **Total** |
> |-----------|-----------|-----------|-----------|-----------|-----------|-----------|
> | **Token usage** | | | | | | |
> | Gemini 2.5 Pro (API) | 26.0M | 13.5M | 25.1M | 4.3M | 14.0M | 82.9M |
> | **Evaluation time (hours)** | | | | | | |
> | Qwen2.5-Omni | 6.3 | 1.4 | 1.4 | 1.4 | 2.1 | 12.7 |

---

> ### Author Response · Authors · 2025-11-26
> **Analysis of SOTA Performance (Gemini 2.5 Pro)**
>
> **Your Concern:**
>
> > (Weakness 2) The paper reports Gemini 2.5 Pro as the strongest model, but offers limited analysis of *why*. A deeper discussion of potential reasons—architecture, data, encoder alignment—would strengthen the contribution.
> >
> > (Question 3) Do the authors have hypotheses on why Gemini 2.5 Pro demonstrates stronger cross-modal consistency and overall competence?
>
> **Response:**
>
> We agree and will expand our analysis. While proprietary details of model architecture, data, and training recipe are unavailable, we can hypothesize based on observable behavior and prior literature:
>
> 1. **Cross-Modal Encoder Alignment:** Gemini 2.5 Pro appears to use tightly coupled encoders for audio, image, and video, leading to more consistent latent alignment than typical open-source pipelines where modalities are pretrained separately.
>
> 2. **Training Data Domain Coverage:** The model shows robustness across language, audio, and visual tasks, suggesting large-scale multimodal pretraining that includes mixed audio–image–video contexts.
>
> 3. **Long-Context Reasoning:** Its stability on multi-step reasoning tasks indicates an architecture optimized for long-context fusion, unlike many open-source models that struggle with multi-hop cross-modal inference.
>
> 4. **Remaining Weaknesses:** Despite these strengths, Gemini still exhibits clear deficits in spatial and temporal reasoning—precisely where OLLM capabilities remain underdeveloped.
>
> **In the new Section 5, We add a dedicated subsection comparing Gemini 2.5 Pro with state-of-the-art (SOTA) models for behavior analysis, and how our finding can inspired model development.**

---

> > ### Comment · Area_Chair_9apV · 2025-11-27
> > **Ethics issue**
> >
> > Dear authors,
> >
> > You are expected to address the flagged ethics issue and provide an explanation or additional materials as soon as possible.
> >
> > Your AC

---

> > > ### Author Response · Authors · 2025-11-27
> > > **Ethics Statement**
> > >
> > > Thanks for your comment. Below is the additional Ethics Statement and we also added this to the paper.
> > >
> > > ---
> > >
> > > ## Ethics Statement
> > > Our study does not involve private or sensitive personal data. All audiovisual samples are obtained from publicly available official sources, including previously published research datasets, content hosted on established open source platforms such as Hugging Face and Kaggle.
> > >
> > > For all newly generated labels and annotations, we perform manual verification to ensure correctness and to remove any potentially inappropriate content. All web-curated data are from publicly accessible and previously published sources without requiring special authentication. All materials are used solely for non-commercial academic research. We do not redistribute copyrighted video or audio; only derived features, annotations, and evaluation results are released.

---

### Official Review · Reviewer_3qp2 · 2025-10-31

**Soundness:** 3
**Presentation:** 1
**Contribution:** 2
**Rating:** 4
**Confidence:** 3

**Summary:**

This paper introduces XModBench, a large-scale tri-modal benchmark (audio, vision, text) designed to evaluate cross-modal consistency in omni-modal large language models (OLLMs). The dataset contains 60K multiple-choice QA pairs rendered in six modality directions (e.g., audio→text, image→audio), covering five task categories: perception, spatial reasoning, temporal reasoning, linguistic understanding, and external knowledge. The authors benchmark several frontier models (e.g., Gemini 2.5 Pro) and report three key findings: (i) OLLMs still struggle with spatial/temporal reasoning, (ii) performance drops significantly when audio replaces text, and (iii) models show strong directional imbalance (e.g., text→vision vs. vision→text). The benchmark aims to serve as a diagnostic tool for measuring modality-invariant reasoning.

**Strengths:**

The paper explicitly targets cross-modal consistency, a dimension often ignored in existing multimodal benchmarks.

Each question instance is rendered across all six modality mappings, enabling controlled comparison and directional analysis.

Large scale and broad coverage.
The dataset includes >60K QA samples spanning 17 subtasks, with balanced modality construction.

Relevance to current model trends.
As many new models claim “omni-modality,” this benchmark fills a timely evaluation gap.

**Weaknesses:**

Table 2 is overloaded and hard to interpret.
The key conclusions (e.g., text→image > audio→text; Gemini has lowest variance) are meaningful, but the table is dense, lacks focused analysis, and could be split into smaller tables aligned with each main claim.

Interesting modality-swap results but no deeper investigation.
The paper observes asymmetric performance (e.g., vision→text vs. text→vision) but does not analyze why. For example:
– Do any models use interleaved multimodal training data?
– Do models with such data show smaller swap gaps?

Dataset quality control is unclear.
The benchmark claims 60K samples but does not report human validation, error rate, or annotation quality checks.

No discussion of answer-option bias.
Some modalities may allow shortcut guessing (e.g., lexical cues in text choices). There is no “noise-input” baseline to rule this out (e.g., Gemini with shuffled / blank modality input).

Lack of analysis for <25% performance cases.
Several settings score worse than random guessing (25% for 4-choice MCQ), but the paper does not explain whether this is due to instruction following, noisy inputs, or poor distractor design.

**Questions:**

Dataset quality
Have you conducted human verification on a subset of the 60K samples? If so, what is the estimated annotation error rate?

Distractor bias
Can models guess correct answers without context? Please provide “no-input” or “noise-input” baselines to quantify answer-option bias.

Modality swap analysis
Do any evaluated models train on interleaved multimodal corpora (e.g., narrated video, audiocaps)? If yes, do they exhibit smaller directional gaps?

Table 2 clarity
Would you consider splitting Table 2 into multiple focused tables (e.g., task competence, disparity, imbalance) to improve readability?

Below-random performance
For conditions where models score <25%, what is the failure mode? Instruction refusal? Systematic misalignment? Poor distractor construction?

Benchmark extensibility
Do you plan to release tools for adding new modality pairs (e.g., text↔3D, audio↔video)?

---

> ### Author Response · Authors · 2025-11-26
> **1. Clarity and Readability of Table 2**
>
> **Your Concern:**
>
> > (Weakness 1) Table 2 is overloaded and difficult to interpret. Although the main conclusions are meaningful, the table is dense, lacks focused analysis, and could be split into smaller, more targeted tables.
>
> **Response:**
>
> Thank you for pointing this out. We agree that the original Table 2 was dense and hard to interpret. **In the revised pdf**, we have **split Table 2 into two separate subtables** for improved readability and align with our main finding in Section 4.2:
>
> 1. a **task-wise accuracy subtable** that summarizes model performance across the five task families, and
>
> 2. a **modality-configuration subtable** that focuses specifically on different modality configuration on the overall dataset.
>
> This separation improves readability and makes it easier for readers to interpret both cross-modal and task-specific behaviors.

---

> ### Author Response · Authors · 2025-11-26
> **2.  Modality-Swap Results and Lack of Deeper Analysis**
>
> **Your Concern:**
>
> > (Weakness 2) The modality-swap results (e.g., V→T vs. T→V) are interesting but not analyzed. Is asymmetry related to whether models were trained on interleaved multimodal corpora? Do such models show smaller directional gaps?
>
> **Response:**
>
> We appreciate the reviewer's insight. Our only concern is the training data of most state-of-the-art multimodal models is largely opaque, which makes it difficult to draw definitive causal conclusions about modality-swap asymmetry. Nevertheless, following your suggestion, we systematically compared models based on all publicly available documentation and reports. From these sources, we can make evidence based observations regarding how **data composition** (e.g., use of interleaved multimodal corpora) and **data-domain coverage** shape model behavior:
>
> 1. **Models with known interleaved training sources show smaller directional gaps.** Public reports indicate that models such as *Qwen-Omni* and Google's *Gemini* series incorporate interleaved multimodal data (e.g., narrated videos, audio–caption pairs, mixed audio–vision corpora). Although the exact ratios remain undisclosed, the presence of such interleaving aligns with the relatively **balanced modality-swap performance** observed in our benchmark.
>
> 2. **Models relying on limited or lightly interleaved open-source data exhibit larger asymmetry, even when built upon strong backbones.** For example, *EchoInk-R1* is trained primarily on open-source multimodal data, where the proportion of interleaved audio–vision instruction data is **significantly smaller** compared with Qwen or Gemini. Despite achieving higher overall accuracy through instruction tuning and reinforcement learning, EchoInk-R1 shows **greater imbalance** in modality-swap results, suggesting that insufficient interleaved multimodality can hinder directional robustness.
>
> 3. **Domain coverage gaps further amplify task-specific weaknesses.** EchoInk-R1's training data lacks sufficient **spatial-vision** content, leading to a sharp performance drop on spatial reasoning tasks relative to Qwen-Omni-2.5, despite their related model families. This indicates that **data-domain imbalance** (not just data volume) disproportionately affects certain task families and contributes to asymmetric modality behavior.
>
>
> ---
> ### **Broader implication and our contribution.**
>
> This opacity of training data is itself a challenge for the community. It further underscores the necessity of our comprehensive benchmark:
>
> - **For future model developers**, only a detailed and diagnostic benchmark such as ours can reveal how data composition (interleaved vs. disjoint) and data-domain imbalance manifest in downstream multimodal reasoning—insights that would otherwise remain hidden without data transparency.
>
> - **For current model builders**, deveoping with internal training data would facilitate clearer analysis and adjustment of data pipelines, enabling more stable multimodal alignment and reducing modality-swap inconsistencies.
>
> Such analysis is not achievable with existing benchmarks, which lack the multimodal invariant and modality-swap design that is necessary to disentangle data-related effects. We add a more detailed discussion in the new Section 5.

---

> ### Author Response · Authors · 2025-11-26
> **3. Dataset Quality Control and Human Verification**
>
> **Your Concern:**
>
> > (Weakness 3) The benchmark includes 60K samples, but the paper does not explain human validation or annotation quality control.
>
> **Response:**
>
> We appreciate the reviewer's concern. The dataset actually underwent multiple rounds of internal iteration and stringent quality control (as described in Section 3.3), during which a large number of problematic cases were removed. We will make this explicit in the revision. Our quality-control process consists of three components:
>
> 1. **Systematic removal of ambiguous or low-quality samples.** We manually filtered cases containing (i) overly similar answer choices such as *singing* versus *speaking* or *splashing water* versus *squishing water*, (ii) low-quality or noisy audio that made temporal counting tasks unreliable, (iii) incorrect TTS outputs in speech-text tasks, and (iv) spatial arrangements requiring fine-grained distinctions that the source data could not support, for example, *cello* versus *viola*.
>
> 2. **Manual screening of all annotations.** Every QA pair produced during data construction was manually inspected to ensure correctness, proper modality grounding, and unambiguous distractor design. Approximately 20-30% of the data were removed during the curation process before reaching the final release. We apologize that the earlier intermediate files were deleted, so we are unable to provide the exact removal ratio.
>
> 3. **Human verification of the finalized benchmark.** But the final quality can be ensured. We conducted a randomized human evaluation on the finalized dataset quality. Human judged correctness on the question sample across all task families and modality, that achieved an accuracy above **88%** (see Table 2), confirming that the benchmark maintains a high standard of annotation quality. Inevitably, some human errors are acceptable and expected—for example, annotators may be unfamiliar with certain musical instruction in the perception tasks, may miscount event repetitions in temporal tasks, or may not recognize specific movies referenced in external-knowledge questions. These cases reflect natural limitations of human prior knowledge rather than flaws in the dataset construction.

---

> ### Author Response · Authors · 2025-11-26
> **4. Answer-Option Bias and Noise-Input Baselines**
>
> **Your Concern:**
>
> > (Weakness 4) The benchmark may contain lexical shortcuts or answer-option biases. A "no-input" or "noise-input" baseline is needed to rule out guessing.
>
> **Response:**
>
> We agree that verifying option bias is important. To address this concern, we conducted a controlled evaluation under the **"no-context"** setting using *Gemini 2.5 Pro*. We removed all task inputs and provided only the original answer options, while explicitly instructing the model to select an answer even in the absence of input context. Since no information was available, the prompt encouraged the model to guess. As shown in Table 2, the resulting accuracy is close to the random-guess rate of approximately 25%.
>
> These findings confirm that models cannot solve tasks using answer-only heuristics and that XModBench does not suffer from option-selection shortcuts. The updated results are incorporated into the revised Table 2. Following the reviewer's request for clearer presentation, several detailed results that previously appeared in Table 2 have been reorganized into the new Table 5 and Appendix B,C for deeper inspect.

---

> ### Author Response · Authors · 2025-11-26
> **5. Below-Random Performance Cases**
>
> **Your Concern:**
>
> > (Weakness 5) Several settings achieve <25% accuracy (below random guessing in 4-choice MCQ). The failure cause is not analyzed.
>
> **Response:**
>
> Thank you for the suggestion. We looked into these cases and found that the lowest scores (around 20%) mostly come from models without interleaved multimodal training, such as *PandaGPT* and *Unified-IO*. During development, we already noticed this and double-checked that all model outputs followed the correct answer format (A, B, C, D). If the model response was unclear or malformed, we used *LLM (i.e. Gemini 2.5 flash lite)* to extract the correct option from the output text. So the reported numbers are accurate.
>
> For these weak models, we believe that random guessing is their actual behavior, since they lack the ability to use cross-modal information. To confirm this, we also ran the **no-input** baselines (as requested earlier), and the results were similarly close to 25% with uniform option choices. This shows that our benchmark does not introduce bias or shortcuts, and that the low performance is due to model limitations.

---

> ### Author Response · Authors · 2025-11-26
> **6. Benchmark Extensibility**
>
> **Your Concern:**
>
> > (Weakness 6) Do the authors plan to release tools for adding new modality pairs (e.g., Text↔3D, Audio↔Video)?
>
> **Response:**
>
> Yes. **We will open-source the complete data generation pipeline and all task-specification tasks, new annotated tri-modality data,**  which allows the community to extend XModBench to new modality pairs. We hope that releasing the full pipeline can encourage more open-source work and help model developers pay attention not only to absolute accuracy, but also to balanced across modalities. We believe this will lead to more reliable and well-grounded multimodal systems in the long run.

---

> ### Comment · Reviewer_3qp2 · 2025-11-27
>
> Thanks the author to address my concern and I change my rating to 6 accordingly.

---

> > ### Author Response · Authors · 2025-11-27
> >
> > Thank you for the positive update! We appreciate your careful review and are grateful that the revisions addressed your concerns.

---

### Official Review · Reviewer_2qMR · 2025-11-01

**Soundness:** 2
**Presentation:** 3
**Contribution:** 3
**Rating:** 4
**Confidence:** 5

**Summary:**

This paper introduces XModBench, a large-scale tri-modal benchmark specifically designed to measure cross-modal consistency in Omni-modal Large Language Models (OLLMs) by systematically covering six cross-modality directions for audio, vision, and text. XModBench comprises over 60,000 multiple-choice questions across five task families and 17 subtasks, enabling diagnostic assessment of task competence, modality disparity, and directional imbalance.

**Strengths:**

- Comprehensive Diagnostic Scope: XModBench provides a large-scale, systematically balanced tri-modal QA benchmark, covering all six modality permutations (audio, vision, text) for both the context and candidate answers. The benchmark covers five diverse task families (perception, spatial, temporal, linguistic, external knowledge), each with multiple subtasks.

- Detailed Empirical Analysis: The authors conduct a detailed empirical analysis of cutting-edge OLLMs, including a performance breakdown by task and modality configuration (Table 2). This analysis effectively identifies the significant lack of capability or competence in current OLLMs within the audio domain.

**Weaknesses:**

- XModBench primarily focuses on isolated cross-modal alignment (e.g., T→V, V→T) and fails to cover true mixed tri-modal capabilities (e.g., Image+Vision+Audio→Text/Image). This combined modality reasoning is arguably the critical differentiator separating OLLMs from Multimodal Large Language Models (MLLMs) and specialized speech models.

-  The data curation shows an over-reliance on GPT-5 as the primary question generation tool. However, the quality assurance (QA) or filtering process for this synthetic data is not clearly elaborated. This risks labeling XModBench as a 'silver' dataset rather than a 'gold' standard, where prioritizing data quality over sheer quantity is paramount.

-  Contextualization against MLLMs: The analysis lacks comparison against the performance of MLLMs focused on traditional ASR or image-to-text (I2T) tasks, such as Qwen-VL or Intern-VL. It remains unclear whether OLLMs maintain a performance advantage in these specific subdomains when compared to these more focused MLLMs.

**Questions:**

same as weakness

---

> ### Author Response · Authors · 2025-11-26
> **1. Tri-Modal Reasoning Coverage**
>
> **Your Concern:**
>
> > (Weakness 1) XModBench focuses primarily on isolated cross-modal alignment (T→V, V→T), and does not evaluate true mixed tri-modal capabilities (Image + Video + Audio → Text/Image), which are critical for diagnosing OLLMs.
>
> **Response:**
>
> We appreciate the reviewer's suggestion and agree that the AV→T setting represents a stronger form of multimodal mixing in an omnimodal benchmark.
>
> Importantly, our A→V and V→A tasks already involve all three modalities, with text serving as the instruction while audio and vision serve as the answer choices directly. This configuration is arguably more challenging than existing OLLM benchmarks (e.g., OmniBench), where multimodal inputs are often less tightly intertwined. Our design requires models to jointly ground audio–visual semantics while following a textual query, thereby probing a deeper level of cross-modal reasoning.
>
> **In Section 4.6, we additionally introduce the AV→T variant and evaluate SOTA models to examine how their behavior changes when richer multimodal context is provided**. The results show that adding audio–vision context has inconsistent and relatively small effects across models: Gemini 1.5 Pro: +2.61, Gemini 2.0 Flash: –5.36 and Gemini 2.5 Pro: +1.10.
>
> (1) These fluctuations are minor compared to the improvements observed when using vision-only as contextual input, supporting our hypothesis that audio and vision often encode overlapping semantics, causing models to rely predominantly on visual cues.
>
> (2) This result also reinforces our findings in Section 4.3 (Modality Disparity Analysis), where audio consistently emerges as the weakest modality among current OLLMs.
>
> While we acknowledge the value of more comprehensive tri-modal evaluation, but the main focus of our work is on **modality inconsistency and modality-invariant** task performance. In this context, the current results already demonstrate that tri-modality remains largely auxiliary for existing models and the extra multi-modality is redundancy without new clear pattern.

---

> ### Author Response · Authors · 2025-11-26
> **2. Use of GPT-5 in Data Generation and Data Quality Assurance**
>
> **Your Concern:**
>
> > (Weakness 2) The benchmark appears to over-rely on GPT-5 for question generation, and the QA/filtering process is not clearly explained. This raises concerns about the dataset being "silver" rather than "gold" quality.
>
> **Response:**
>
> We appreciate the opportunity to clarify this point. **GPT-5 is used solely to rephrase the template of each question, without adding external knowledge or altering its semantic meaning**. XModBench employs a **template-based** generation pipeline, where all questions are instantiated directly from our tri-modal annotations (as illustrated in Figure 3). GPT-5 is applied only to refine grammar, fluency, and instruction clarity—common limitations of template-generated text.
>
> To ensure faithfulness, we additionally perform manual verification on all rewritten questions.
>
> To address concerns about dataset quality, we emphasize the following aspects of our pipeline:
>
> - **Deterministic generation:** Questions are generated using predefined templates applied to our manually verified annotations, ensuring consistency and controllability.
> - **Human verification:** All annotations undergo rigorous validation before template application (detailed in Section 3.3).

---

> ### Author Response · Authors · 2025-11-26
> **3. Comparison Against MLLMs for VL / ASR Tasks**
>
> **Your Concern:**
>
> > (Weakness 3) The analysis lacks contextualization against MLLMs specialized for VL or ASR tasks (e.g., Qwen-VL, InternVL). It is unclear whether OLLMs maintain a performance advantage in these subdomains.
>
> **Response:**
>
> We agree that including this comparison further strengthens the contribution of our work.
>
> For reviewer reference, we report the overall V→T and T→V performance of Qwen2.5-VL and InternVL3.5-VL in Table 1, with comparison to Qwen2.5-Omni on both the V-T subset and the full 6-modality configuration.
>
> We have incorporated these results into the updated Table 2 of the paper.
>
> We observe the following:
>
> (1) **Qwen2.5-VL exhibits similar behavior to Qwen2.5-Omni on vision–text tasks.** Both models show weaknesses in temporal and spatial reasoning, while achieving competitive results in perception, linguistic, and external knowledge tasks. The slightly higher task-specific accuracy of the VL model is mainly due to lower performance on audio-related tasks—consistent with the modality disparity analysis in Section 4.3.
>
> (2) **Directional imbalance persists in text–vision tasks.** For both models, performance on T→V and V→T tasks is tiered, which means the directional imbalances also exist on the VLM, that V→T (74.7) notably outperforming T→V (60.1), resulting in a gap of 14.6.
>
> (3) **InternVL3.5-VL shows an even larger directional imbalance.** Its accuracy on V→T (73.7) substantially exceeds that on T→V (49.7), leading to a gap of 24.0—almost twice the discrepancy observed in Qwen2.5-VL.
>
> These results reinforce our main conclusion that **modality-directional imbalance is a model-agnostic phenomenon**, consistently appearing across different VLM families and further validating the diagnostic value of XModBench.
>
> For the ASR task, we noticed that existing state-of-the-art ASR systems (e.g., Samba-ASR) are not applicable to our benchmark. These models are designed exclusively for *speech recognition* and operate within a narrow audio domain, and the Gemini-2.5-Omni has served as an SOTA API-based audio reasoning model.
>
> **Table 1:** Result of QwenVL and InternVL on the overall V→T and T→V modality configuration and comparison to Qwen2.5-Omni in V-T subset and on 6 modality configuration.
>
> | **Model** | **Accuracy** |  **on 5 Task**  |  **Families**|  |  | **Modality** | **Configuration** | **Avg.** |
> |-----------|---------|---------|---------|---------|---------|---------|---------|----------|
> |           | **Perc.** | **Spat.** | **Temp.** | **Ling.** | **Knwl.** | **T → V** | **V → T** |          |
> | Qwen2.5-VL | 91.3 | 51.4 | 40.9 | 84.1 | 77.2 | 60.1 | 74.7 | 67.4 |
> | Intern3.5-VL | 87.2 | 42.7 | 41.4 | 75.0 | 68.7 | 49.7 | 73.7 | 61.7 |
> | Qwen2.5-Omni (V-T task only) | 91.6 | 51.1 | 38.2 | 76.1 | 82.8 | 59.6 | 76.3 | 67.9 |
> | Qwen2.5-Omni | 75.5 | 38.4 | 32.3 | 74.1 | 72.8 | 59.6 | 76.3 | 58.6 |

---

### Author Response · Authors · 2025-11-26
**We thank all reviewers for their thoughtful and constructive feedback.**

We sincerely thank the new area chairs for their time and effort in handling our submission. We thank all reviewers (2qMR, 3qp2, a37s, DEkV) for their thoughtful and constructive feedback. We are encouraged that the reviewers unanimously recognize the novelty, comprehensiveness, and diagnostic value of XModBench.

Below, we summarize the consensus on strengths and address the common questions raised across multiple reviews.

## 1. Summary of Acknowledged Strengths

**Critical & Novel Research Focus:** Reviewers commended the paper for tackling the underexplored problem of Cross-Modal Consistency in Omni-modal LLMs (OLLMs). 3qp2 and a37s highlighted that this fills a significant evaluation gap, while DEkV noted the "insightful diagnostics" that reveal hidden biases often missed by standard accuracy metrics.

**Comprehensive & Systematic Design:** The rigorous construction was widely appreciated. Reviewers praised the large scale (>60K samples) and diverse coverage (5 task families, 17 subtasks) (2qMR, 3qp2, a37s). The unique design covering all 6 modality permutations was cited as a major strength for enabling controlled comparisons (2qMR, 3qp2, DEkV).

**High-Quality Curation:** a37s specifically praised the "High-Quality Curation" and the "human in-the-loop verification," noting that our rigorous process addresses major concerns about synthetic data ambiguity.

## 2. Response to Common Concerns

We have addressed the common concerns raised by the reviewers through targeted clarifications, detailed analyses and additional experiments. Point-by-point responses to all comments are provided in the separate threads.

**Concern 1: Deeper Analysis of Model Behaviors (a37s, 3qp2)**

**Response:** As suggested by the reviewers, we have added a discussion (Section 5) explaining the underlying causes of the weaknesses revealed by our benchmark—specifically, modality imbalance and disparity. We attribute these issues primarily to how interleaved multimodal data are used during training. Furthermore, we extend this analysis to data-domain coverage and its connection to post-training by comparing behavioral differences across models (e.g., Qwen-2.5-omni and EchoInk).

XModBench serves not only to report performance gaps but as a diagnostic tool for OLLM developers. Our findings indicate that optimized interleaved multimodal training and targeted domain coverage can substantially reduce modality disparities.

**Concern 2: Data Quality & Bias Control (2qMR, 3qp2, DEkV)**

**Response:** We clarified that GPT-5 is used strictly for grammatical rephrasing of deterministic templates. Our quality-control process includes the systematic removal of ambiguous samples, manual screening of annotations, and multi-round human verification of the finalized benchmark. This ensures high data quality and prevents potential information loss during task design.

**Concern 3: Scope & Baselines (2qMR, DEkV)**

**Response:**

- **MLLM Comparison:** We added Qwen2.5-VL and InternVL3.5 to Table 2. They exhibit even larger directional imbalance (e.g., Δ > 20% between V → T and T → V) than OLLMs, confirming that modality asymmetry is a fundamental, model-agnostic issue.
- **No Shortcut Bias:** A new "No-input" baseline (guessing without context) yields ≈ 25% accuracy (random chance), proving that models cannot solve questions via option shortcuts alone.
- **Tri-Modal task:** The new experiments show mixed AV → T inputs provide negligible gains over Vision-only. Human benchmarks confirm text queries preserve sufficient semantics;

**Concern 4. Usability & Presentation (a37s, 3qp2)**

**Response:**

- **XModBench-Lite:** We are releasing a balanced 6K sample subset to enable low-cost, rapid diagnostics. As the Lite version yields results highly correlated with the full benchmark, it allows developers to choose their preferred scale without compromising evaluation reliability.
- **Cost & Clarity:** We added a Cost Analysis table (API tokens/GPU hours)
- **Presentation**: We split the dense Table 2 into focused subtables for better readability.

## 3. Summary of Revisions

- **New Discussion Section:** We provide a deeper connection between model weaknesses and training factors (data domain, interleaving format) to guide the development of more robust OLLMs.
- **Baseline Extension:** Added the "No-Input" baseline and additional VLM comparisons.
- **Dataset Extension and usability:** Added details for the "XModBench-Lite" subset and the evaluation cost.
- **Ethics Statement:** We clarify that our study does not involve private or sensitive personal data. All audiovisual samples are obtained from publicly available sources. All materials are used solely for non-commercial academic research.

We believe these clarifications and new experiments strengthen the paper and its contribution to the community. We invite the reviewers to examine our detailed point-by-point responses below and look forward to further discussion.

---

### Meta-Review · Area_Chair_t8zk · 2026-01-15

**Summary:**

This paper proposes a benchmark to evaluate the modality-invariant reasoning and modality-specific bias of omni-modal large language models (OLLMs). The proposed XModBench contains 60K multiple-choice questions across five task families and systematically covers all six cross-modality directions, enabling diagnosis of task competence, modality disparity, and directional imbalance. Evaluations on several OLLMs reveal that even the best-performing model, Gemini 2.5 pro, still suffers from modality disparity and directional imbalance. The authors also provided analysis and insights on why they fail and how we can improve OLLMs.

**Reviewer Concerns:**

- Data quality control has not been discussed. Data curation relies on GPT-5 without human justification. Lack of analysis regarding shortcut bias for multiple-choice questions.
- Lack of depth in discussing the reasons behind the modality imbalance in modality-swap tasks and poor performance on some models. Insufficient discussion of other confounding factors, such as information loss in some modality.
- The six tasks in the benchmark enumerate all two-modality combinations for input and output modalities but do not cover tri-modality, such as A+V to T and V+T to A.
- Lack of evaluation of MLLMs on the proposed benchmark.
- Lack of evaluation cost analysis.
- It is not clear how the analysis can provide insights for improving OLLMs.

**Reviewer Scores:**

- The initial review ratings are 4, 4, 4, 6, with corresponding confidence of 5, 3, 3, 4.
- After the rebuttal, a reviewer of 4 (confidence 3) responded and raised the rating to 6, making this paper a borderline submission.
- I have carefully checked the response and discussion. Several major concerns have been addressed by the rebuttal.
- That being said, the responses to (1) the lack of tri-modal tasks and (2) practically feasible training strategies for improving cross-modal consistency do not address the concerns directly. For example, a textual prompt cannot fully represent an additional input modality, as the context is still in a single modality.
- Although the benchmark is novel, the observations and insights (modality disparity and directional imbalance) are widely known and thus less novel to some extent.
- I still feel that developing new benchmarks to study multi-modal consistency in OLLMs should be encouraged. Hence, I lean to accept the paper.

---

### Decision · Program_Chairs · 2026-01-26

Accept (Poster)